# Structured inhibitory activity dynamics in new virtual environments

**Moises Arriaga, Edward B Han***

Department of Neuroscience, Washington University School of Medicine, St. Louis, United States

**Abstract** Inhibition plays a powerful role in regulating network excitation and plasticity; however, the activity of defined interneuron types during spatial exploration remain poorly understood. Using two-photon calcium imaging, we recorded hippocampal CA1 somatostatin- and parvalbumin-expressing interneurons as mice performed a goal-directed spatial navigation task in new visual virtual reality (VR) contexts. Activity in both interneuron classes was strongly suppressed but recovered as animals learned to adapt the previously learned task to the new spatial context. Surprisingly, although there was a range of activity suppression across the population, individual somatostatin-expressing interneurons showed consistent levels of activity modulation across exposure to multiple novel environments, suggesting context-independent, stable network roles during spatial exploration. This work reveals population-level temporally dynamic interneuron activity in new environments, within which each interneuron shows stable and consistent activity modulation.

DOI: https://doi.org/10.7554/eLife.47611.001

## Introduction

Excitation is balanced by inhibition in neuronal networks (*Andersen et al., 1963*). In cortical circuits, feedforward inhibition is rapidly and robustly recruited by excitatory inputs, while pyramidal neuron firing elicits feedback inhibition to further dampen excitability (*Alle et al., 2001*; *Lamsa et al., 2005*; *Pouille and Scanziani, 2001*; *Pouille and Scanziani, 2004*). Furthermore, inhibition strongly controls synaptic plasticity, a putative cellular mechanism of learning (*Bliss and Lomo, 1973*; *Whitlock et al., 2006*). Intact inhibition limits potentiation to relatively low levels while pharmacologically blocking inhibition facilitates both the induction and magnitude of potentiation (*Artola and Singer, 1987*; *Bear et al., 1992*; *Steward et al., 1990*; *Wigström and Gustafsson, 1983*). Thus, inhibition suppression is a potential mechanism for enhancing learning by favoring synaptic plasticity in excitatory neurons.

Notably, inhibition can be strongly modulated in vivo in freely moving rodents. CA1 fast-spiking putative interneurons are suppressed when rats explore a novel spatial environment (*Frank et al., 2004*; *Nitz and McNaughton, 2004*; *Wilson and McNaughton, 1993*). Learning new food locations in a familiar environment dynamically modulated fast-spiking interneuron activity and altered the associations of these interneurons with pyramidal cell ensembles (*Dupret et al., 2013*). Hippocampal CA3 parvalbumin-expressing interneurons (PV-ints) have decreased network connectivity during the initial learning of the Morris Water Maze, but this connectivity increases with task performance, a modulation that is is necessary for learning (*Donato et al., 2013*; *Ruediger et al., 2011*). Furthermore, numerous studies have shown that suppression of somatostatin- and/or parvalbumin-expressing interneurons (SOM-ints and PV-ints) is necessary for certain types of cortical and amygdalar learning. This suppression is often triggered by disinhibitory inputs from interneurons that preferentially target other interneurons for inhibition, including vasoactive intestinal peptide-expressing interneurons (VIP-ints) (*Gentet et al., 2012*; *Karnani et al., 2016*; *Lee et al., 2013*; *Letzkus et al.,*

***For correspondence:**
ehan23@wustl.edu

**Competing interests:** The authors declare that no competing interests exist.

*2011*; *Makino and Komiyama, 2015*; *Mardinly et al., 2016*; *Pi et al., 2013*; *Turi et al., 2019*; *Wolff et al., 2014*). Finally, the activity or plasticity of numerous other interneuron cell-types has been implicated in controlling animal behaviors (*Basu et al., 2016*; *Hartzell et al., 2018*). Together, this work demonstrates the dynamic nature of inhibition during spatial exploration and learning, and identifies the importance of inhibition suppression in certain types of learning.

Our understanding of in vivo inhibitory activity in the hippocampus is primarily driven by recordings of soma-targeting fast-spiking interneurons (likely PV-ints) since their distinctive firing characteristics make them relatively identifiable in extracellular electrophysiology recordings (*Frank et al., 2004*; *Klausberger et al., 2004*; *Nitz and McNaughton, 2004*; *Wilson and McNaughton, 1993*). These studies found transient suppression in firing of fast-spiking units during exploration of novel environments, consistent with a model in which decreased inhibition is permissive for excitatory plasticity and downstream learning. Interestingly, studies using calcium imaging in head-fixed animals in visual virtual reality (VR) contexts have found conflicting results with freely moving animals, with either no change in PV-int calcium activity in new virtual environments or a decrease in somatic calcium activity coupled with increased putative axonal calcium fluorescence, both suggesting no decrease in perisomatic inhibition (*Hainmueller and Bartos, 2018*; *Sheffield et al., 2017*).

SOM-ints, although far less understood, are of interest because they selectively innervate the dendrites of pyramidal neurons and can directly control dendritic excitability. Dendritic spikes, typically characterized by calcium entry through $Ca^{2+}$ channels or NMDA receptors, generate burst firing of neurons and can mediate long-term plasticity, place field formation, and learning (*Bittner et al., 2015*; *Bittner et al., 2017*; *Cichon and Gan, 2015*; *Golding et al., 2002*; *Larkum et al., 1999*). Formation of place fields is associated with dendritic spikes that occur during transient periods of SOM-int activity suppression in novel environments (*Sheffield et al., 2017*). In contrast, SOM-int activation rather than suppression is required for fear learning, both in CA1 or in the dentate gyrus (*Lovett-Barron et al., 2014*; *Stefanelli et al., 2016*).

Interneuron activity may be dynamically modulated in new environments or during learning, but the stability of these activity dynamics in individual interneurons across time remain unknown. This is partially a technical issue as extracellular electrode recordings are typically stable for a few hours and have limited ability to identify interneurons, making the longitudinal recording of single interneuron activity difficult. Many inhibitory interneurons are strongly influenced by pyramidal neurons, driven in a feedforward and/or feedback manner. Thus, if activated pyramidal neuron ensembles are stochastic, as is the case for place cells in different environments, then stochastic ensembles of strongly activated interneurons should result. An alternate possibility is that individual interneurons play consistent and reproducible roles in a context-independent manner, reflecting an underlying structure that determines how interneurons regulate the pyramidal network. While inhibitory neurons are composed of multiple cell-types playing distinct network roles (*Klausberger and Somogyi, 2008*; *Pelkey et al., 2017*; *Wamsley and Fishell, 2017*), little is understood about the functional specialization of interneurons *within* a defined cell-type. Previous work from our lab and others have found functional diversity within the same putative interneuronal cell-types, with individual neurons being consistently activated or inhibited by locomotion (*Arriaga and Han, 2017*; *Garcia-Junco-Clemente et al., 2019*). Here we investigated whether learning to adapt a previously learned task to a new spatial context could similarly reveal the 'set' functional properties of specific interneurons.

We examined the activity dynamics of PV- and SOM-ints using calcium imaging over multiple days of exposure to initially novel environments as animals performed a goal-directed spatial navigation task. We found that PV- and SOM-int activity was strongly suppressed specifically during the initial exploration of new virtual worlds, with activity returning to baseline levels as animals learned to adapt the task to the new context. In contrast, for animals where the recovery of task performance is blocked (static visual scene with no task), SOM-int activity remained persistently suppressed for days, suggesting that the recovery of interneuron activity is tied to recovery of task performance, rather than to either habituation to the context switch or familiarity with the new visual environment. Surprisingly, suppressed interneuron activity triggered by context changes showed a defined population structure across both new environments and the 'No Task' condition in SOM-ints. Each interneuron exhibited consistent activity suppression, with high correlation of suppression across multiple novel contexts as well as the 'No Task' condition. These data reveal interneuron

activity suppression during spatial exploration in new contexts, as well as a functional inhibitory network structure that may route the encoding of information within the pyramidal network.

## Results

### Virtual reality behavior

We used two-photon calcium imaging to stably record from neurons over weeks to study the activity dynamics of the same cells in a goal-directed, spatial navigation task across different virtual environments. We initially trained water-scheduled mice to run to alternating ends of a virtual visual track to receive water rewards, using their movement on a floating spherical treadmill (styrofoam ball) to control their movement in VR (*Arriaga and Han, 2017*; *Dombeck et al., 2010*; *Harvey et al., 2009*). Note that mice need to physically rotate and run on the ball in order to turn and maneuvre through VR, making this an internally consistent and continuous virtual world that contrasts with VR tasks requiring unidirectional movement on a track with either 'teleportation' back to the starting point after each trial (*Hainmueller and Bartos, 2018*; *Sato et al., 2017*; *Sheffield et al., 2017*) or an infinitely repeating corridor (*Gauthier and Tank, 2018*).

We used this task to study interneuronal activity dynamics as animals learned to adapt this previously learned behavior to new virtual contexts. We ran animals for 7 min in the well-trained familiar environment (Fam), immediately switched mice to a new VR environment for 14 min (New), and then switched them back for another 7-min session in the original familiar environment (Fam') (schematic, *Figure 1A*). The task was identical in familiar and New epochs but the visual environment was different, with changes to the walls, track, and distal landmarks. We repeated this protocol over 5 days, with the same New environment each time.

We characterized behavioral performance in novel contexts using a cohort of mice, a subset of which was used for SOM- and PV-int imaging (the remainder used for experiments not discussed here). One quantification of task performance was the number of rewards per minute (rew/min) in Fam, New, and Fam' epochs. Upon initial exposure to the New virtual world, animal behavior was dramatically altered. Performance in New was significantly worse on Days 1, 2, and 4 compared to $Fam_{Ave}$ (the average performance in the flanking Fam and Fam' epochs) (*Figure 1C*, performance in New normalized to $Fam_{Ave}$). This impairment was largest on Day 1 and gradually decreased over the next four days of exposure to the same 'New' world, indicating that animals learned to adapt previously learned behavior to the new context.

The recovery of task performance over days in New could be due to context-specific adaptation of the task (which is likely to be hippocampal-dependent) or the result of a more general context-independent strategy. One example of the latter is the identification of a virtual corridor and navigation to the end of this corridor, without regard for the specific visual context (either Fam or New), because rewards are given at the end zones of the track. This strategy is intuitive in the real world; once animals are trained to run to alternating ends of a track, they can perform the task in new tracks immediately in a non-hippocampal-dependent manner (*Kim and Frank, 2009*). Similarly in a task where mice run forward on a treadmill (constrained to 1D movement), move down a VR track for reward and then are teleported back to the beginning for each trial, task performance recovers within a few minutes of exposure to a new track (*Sheffield et al., 2017*). In marked contrast, recovery of behavioral performance occurs over days in our task.

Furthermore, if mice rapidly identify virtual corridors, we would expect to see early indications of track location awareness in New. In familiar environments, animals typically slow down before entering the end zone (marked by a period of deceleration beginning several seconds prior the end zone) in anticipation of receiving reward and consuming water (*Gauthier and Tank, 2018*). However, in new environments, animals initially did not decelerate as they approached the end zone, but over 5 days of repeated exposure to New, they decelerated more and more, approaching the levels of deceleration seen in $Fam_{Ave}$ (*Figure 1D*, Day 1, p<0.001, t = 5.1; Day 2, 3, and 4, p<0.05, Day 2 t = 3.44, Day 3 t = 3.48, Day 4 t = 2.8). Similarly, animals in familiar environments generally begin licking within a 1 s window centered on reward delivery (which we define as 'correct' licks); in comparison, mice lick with higher frequency outside of this window in New (*Figure 1—figure supplement 1A*, Day 1, p<0.01, t = −4.82; Day 2, p<0.05, t = −3.13), with their behavior improving over repeated training. We characterized other behavioral metrics that show a similar pattern of initial

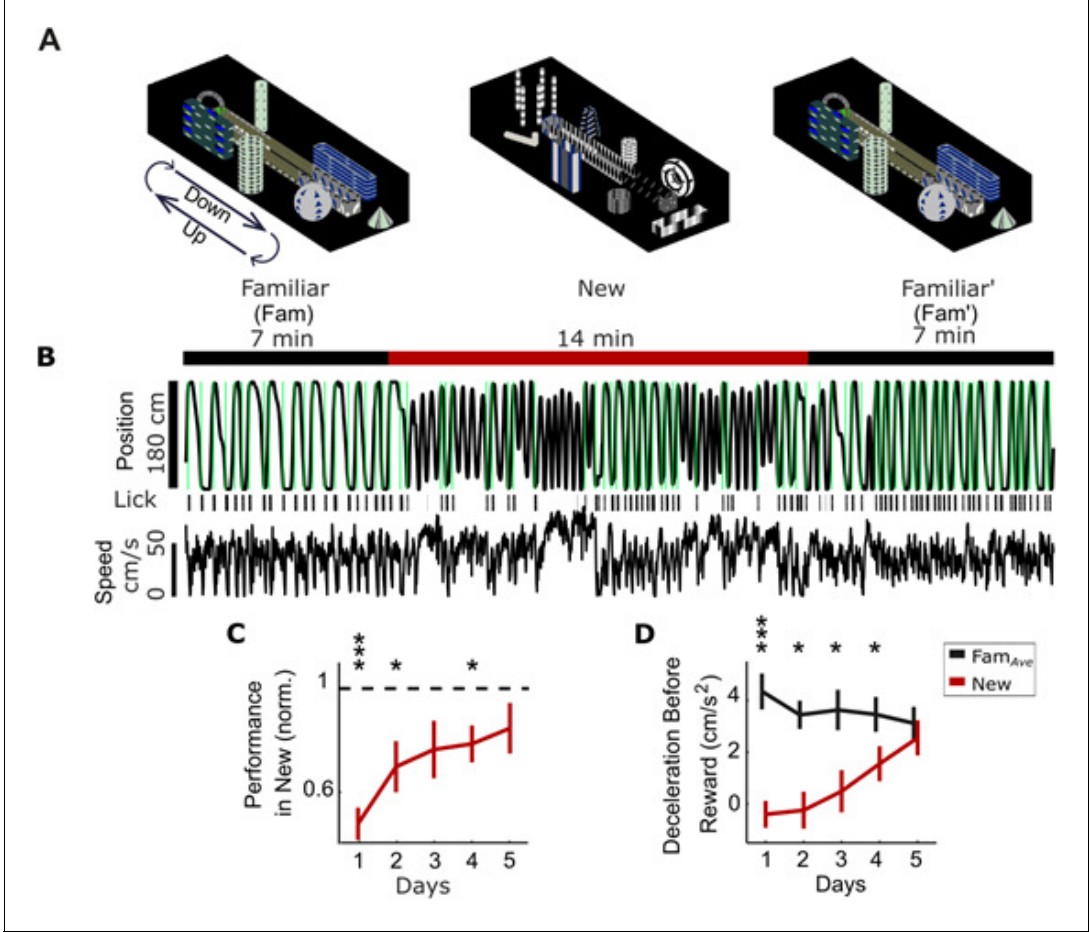

**Figure 1.** Behavior in new visual virtual reality (VR) environments. (**A**) Head-fixed mice run to alternating ends of the VR track by controlling the movement of a floating spherical treadmill (Styrofoam ball). Mice run forward on the ball to traverse the track and rotate the ball to turn around in VR. Animals spend 7 min in a familiar environment (Fam), which is instantaneously replaced with a new environment (New) for 14 min, before returning to the same familiar environment (Fam'). The task is the same but the visual scene differs in the two environments. (**B**) Example mouse position in VR shows running to alternating ends of track with water rewards (green) in Fam, with worse performance in New. Lick bouts (black bars) are tracked with an electronic sensor on the lick tube. Ball speed shows similar magnitude in New and Fam environments. (**C**) Behavioral performance is initially impaired in New (rewards/min in New normalized to $Fam_{Ave}$, the average performance in flanking Fam and Fam' epochs) but improves over time. (**D**) Mice slow down prior to reward in the familiar environments, measured as deceleration in the 3 s window before reward. Deceleration before reward is initially lower in New but increases over days, suggesting anticipation of reward sites (N = 14 mice, *p<0.05, **p<0.01, ***p<0.001 by one-sample t-test with Bonferroni-Holm Correction (**C**) or paired sample t-test (**D**)).

DOI: https://doi.org/10.7554/eLife.47611.002

The following source data and figure supplements are available for figure 1:

**Source data 1.** Statistical tests and results for *Figure 1*.
DOI: https://doi.org/10.7554/eLife.47611.005
**Figure supplement 1.** Behavior metrics in New world.
DOI: https://doi.org/10.7554/eLife.47611.003
**Figure supplement 2.** No track location preference in New.
DOI: https://doi.org/10.7554/eLife.47611.004

impairment and recovery (*Figure 1—figure supplement 1B,C,D*). Other measures of behavior did not change, such as percent of time stopped or average speed (*Figure 1—figure supplement 1E, F*).

If mice employ a strategy of using a virtual corridor as a cue for finding the reward zone, behavioral recovery should be accelerated due to improved cue recognition when exposed to another new environment. Alternatively, if behavioral recovery is context-dependent rather than cued, the

time course of recovery should be similar for a second new environment. We put a subset of mice through a second remapping protocol in which they were exposed to another distinct and novel visual virtual environment, labeled 'New 2', with the original novel environment now labeled 'New 1'. We found that the time course of behavioral recovery was the same for both new environments (*Figure 1—figure supplement 1G*), suggesting context-dependence. We also confirmed there was no significant difference in behavioral performance (using non-normalized rew/min) between the flanking Fam and Fam' epochs, indicating that time-dependent effects such as satiation or fatigue were not responsible for task impairment in the intervening New epoch (*Figure 1—figure supplement 1H*).

While the exact nature of the learning that takes place in new environments is not clear, these data are consistent with mice adapting this previously learned task to new environments by learning to navigate the context-specific sequence of visual cues that lead to alternating end zones in the new world. Furthermore, our data do not support the hypothesis that mice immediately recognize the significance of a virtual 'corridor' and use this strategy to navigate to track ends. However, we note that the recovery of behavioral performance in new environments is not an unambiguous measure of learning, as performance is affected by other factors such as surprise at the context switch and running speed.

One such confound that could inadvertently alter mouse behavior is the brightness of the VR worlds, with bright areas potentially inhibiting movement. We designed the VR environments to be as dim as possible (both through minimal projector brightness and an additional dimming film applied to the rear projection screen) with approximately equivalent brightness across worlds. Even the brightest VR features (which constitute a small fraction of the entire visual scene) were 3 cd/$m^2$, which is within the mesoscopic range for visual function at which mice are frequently behaviorally active (*Denman et al., 2018*; *Schmucker et al., 2005*). Furthermore, we looked for signs of localized inhibition of locomotion by calculating the occupancy time in spatial bins of the track for the Fam, New, and Fam' epochs. Mice spend most time in the end zones (where they stop to lick their water rewards) with no obvious indication of preferential areas of stopped or slowed locomotion (*Figure 1—figure supplement 2*), suggesting that VR brightness was not a significant factor in mouse behavior in our experiments.

## Characterization of neuronal calcium activity in novel virtual environments

To investigate in vivo interneuronal activity dynamics during exposure to novel environments, we used two-photon imaging of neuronal calcium activity during a spatial navigation task in visual virtual reality (*Arriaga and Han, 2017*). We used an electric tunable lens to image a 3-D volume of mouse hippocampal dorsal CA1 by capturing sequential imaging frames along the *z*-axis moving from *stratum pyramidale* through *oriens*, over four to six planes at a frame rate of 5.2–7.8 Hz per plane. Cre-dependent AAV1-Syn-Flex-GCaMP6f was injected into Cre$^+$ transgenic mice to drive a genetically encoded calcium sensor specifically in SOM$^+$ or PV$^+$ hippocampal interneurons. Calcium activity can be taken as a proxy for neuronal activity as multiple studies using simultaneous in vivo imaging and cell-attached patch electrophysiology on the same neurons have found strong correlation between spiking and calcium signals (*Chen et al., 2013*; *Dana et al., 2016*). We, and others, have measured the activity dynamics and coding properties of hippocampal neurons in visual virtual environments and found significant similarity between VR and real world behavior in several aspects of function such as place coding, direction-specificity, place cell remapping in novel environments, and interneuron activity correlation and anti-correlation with locomotion (*Arriaga and Han, 2017*; *Gauthier and Tank, 2018*; *Hainmueller and Bartos, 2018*; *Harvey et al., 2009*; *Sheffield et al., 2017*).

## Interneuron activity suppression in a New environment

To investigate the functional activity dynamics of SOM-ints during spatial exploration, we recorded calcium activity from the same cells while animals performed the VR track-running task in Fam, New, and Fam' over 5 days. SOM-int neuronal activity, measured as ΔF/F, was strongly suppressed upon transition into New (ΔF/F of 6 sample cells from one imaging plane of one mouse, *Figure 2A*; *Video 1* shows activity suppression in another set of SOM-ints). Individual neurons were differentially suppressed with some being relatively unaffected. On returning to Familiar after New in Fam',

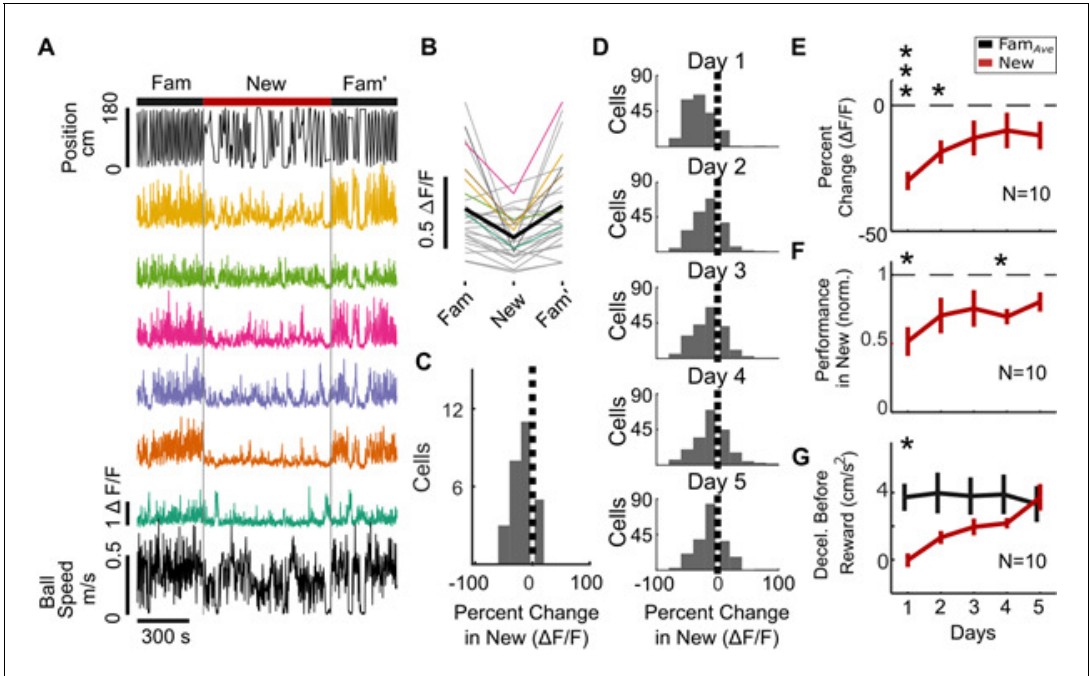

**Figure 2.** SOM$^+$ interneuron (SOM-int) activity suppression in new environments. (A–C) Example data from individual mouse (SOM 1). (A) Top, position in VR track of example mouse. Middle, ΔF/F of sample SOM-ints showing activity suppression in New. (B) Mean ΔF/F of all cells from example mouse on Day 1 of New exposure in each environment (gray), mean ΔF/F of example cells from (A) in corresponding colors, with mean of all cells (black). (C) Histogram of percent change in ΔF/F of SOM-ints shown in panel (B) in New relative to Fam$_{Ave}$ on Day 1. (D) Activity suppression in New decreases with exposure over days (cells from all mice). (E) SOM-int activity is initially suppressed but recovers over days of exposure to New. (F) Performance in New world increases over days. (G) Mice increasingly slow down prior to reward in New. (N = 10, n = 209 cells; n.s. p>0.05, *p<0.05, **p<0.01, ***p<0.001 by paired sample t-test or one-sample t-test with Bonferroni-Holm corrections).

DOI: https://doi.org/10.7554/eLife.47611.006

The following source data and figure supplements are available for figure 2:

**Source data 1.** Statistical tests and results for *Figure 2*.

DOI: https://doi.org/10.7554/eLife.47611.011

**Figure supplement 1.** SOM-int activity suppression in multiple example animals.

DOI: https://doi.org/10.7554/eLife.47611.007

**Figure supplement 2.** SOM-int activity suppression over 5 days of remapping into New.

DOI: https://doi.org/10.7554/eLife.47611.008

**Figure supplement 3.** Broad SOM-int firing fields in Fam and New on Day 1.

DOI: https://doi.org/10.7554/eLife.47611.009

**Figure supplement 4.** Suppression of SOM-int neurite activity.

DOI: https://doi.org/10.7554/eLife.47611.010

calcium activity rapidly recovered, as did behavioral performance (example cells and behavior from an additional three animals in *Figure 2—figure supplement 1*). Similar results can be seen in all cells from this animal (*Figure 2B*). We quantified the neuronal activity of all cells in this example animal as mean ΔF/F and compared this activity across Fam, New, and Fam'. Activity suppression is calculated as the percent difference of mean ΔF/F for each cell between New and Fam$_{Ave}$ using the formula $Percent\ Change = \frac{\Delta F/F_{New} - \Delta F/F_{Fam}}{\Delta F/F_{Fam}} \times 100$. The histogram of percent differences for each cell in the sample mouse shows a distribution of cells that are suppressed in New (*Figure 2C*). The calcium activity from another sample mouse over 5 days of exposure to New follows a similar pattern of activity dynamics (*Figure 2—figure supplement 2*). Across all animals, suppression histograms of SOM activity over the 5-day protocol show a large initial suppression in New that diminishes over days of exposure (*Figure 2D*).

We also examined the spatial correlates of SOM-int activity in Fam, New, and Fam' during Day 1 exposure. We typically observed broad firing fields in Fam and Fam', while activity in New was

**Video 1.** Activity suppression of SOM-ints in New world. GCaMP6f fluorescence in SOM-ints in mouse running in Fam, New (marked 'New World' in movie, 1st day of exposure), and Fam'. Field of view is ~170 μm x ~ 90 μm. Movie is binned into 2 s frames and played at 57.9X speed (original movie is 29.8 min long).
DOI: https://doi.org/10.7554/eLife.47611.012

similarly broad, with the magnitude of activity being reduced in comparison to familiar environments (*Figure 2—figure supplement 3*). Occasional SOM-ints showed firing fields restricted to end zones, representing interneurons activated during immobility (*Arriaga and Han, 2017*).

To quantify this suppression over time, we calculated a percent difference for each mouse by averaging all cells per mouse and then calculated a grand mean for all mice on each day (*Figure 2E*). Indeed, SOM-int activity exhibited significant suppression in New that gradually decreased over days (Day 1, p<0.001, t = −8.70; Day 2, p<0.05, t = −4.06). This decrease in suppression paralleled the increase in behavioral performance in New, as normalized to the average performance in Fam and Fam' (*Figure 2F*, performance worse in New on Days 1 and 4, p<0.05, Day 1 t = −4.34, Day 4 t = −5.57). (Note that metrics in *Figure 2* are from ten SOM-cre mice, a partially overlapping set of the larger behavioral cohort of 14 mice in *Figure 1*). We also found decreased deceleration before rewards in New, which recovered with continued exposure to a New environment (*Figure 2G*, Day 1, p<0.05, t = 3.73). While we used other behaviors such as licking to characterize behavior in *Figure 1—figure supplement 1*, licking behavior was highly variable across animals, leading us to focus on task performance and deceleration before rewards for characterizing behavior in New. These data show, on average, strong suppression of SOM-int upon exposure to a New environment, with recovery of activity over repeated exposures. At the same time, behavioral performance was initially impaired and then increased over time.

Next, we investigated whether SOM-int somatic activity dynamics correlated with activity from nearby SOM-int neurites (axons and dendrites). A previous report found decreased PV-int somatic calcium levels with simultaneous elevation in neuritic calcium, raising the possibility of differential regulation of soma and axon activity (*Sheffield et al., 2017*). By analyzing regions of interest in the *statum oriens* that included neurites and excluded somata, we found that SOM-int neurite activity was similarly suppressed (*Figure 2—figure supplement 4*).

Our data show strong initial inhibition of SOM-int activity in new virtual environments, yet, at the same time, behavior is altered in new environments. Could decreased interneuronal activity directly result from changes in behavior rather than from network reorganization due to spatial exploration? For example, many SOM-ints have activity that is positively correlated with locomotion (*Arriaga and Han, 2017*; *Turi et al., 2019*), making it possible that decreased interneuron activity is due to decreased locomotion in New. Although average locomotion was the same in New and Fam$_{Ave}$ (*Figure 1—figure supplement 1F*), we more closely investigated the relationship between neuronal activity and several behavioral variables, such as locomotion (including speed and acceleration), rewards, and location. As a first pass, we individually correlated these variables with cellular activity and found that, in familiar environments, locomotion-related variables had relatively high correlation with activity, on average. In new environments, these correlations were much lower, indicating a disruption in the relationship between behavioral variables and neuronal activity in new environments (*Figure 3—figure supplement 1*). Similar to the recovery of calcium activity and behavioral performance over time, correlation between behavior and SOM-int activity also recovered with increasing experience in New.

To explore the relationship between behavior and SOM-int activity more thoroughly and quantitatively, we turned to computational modeling using general linear models (GLM) to predict each cell's fluorescence based on behavior. Models were trained using fluorescence data from Fam and fit using forward and rotational components of ball speed, the timing of rewards, and position and speed in the VR environment. Modeled ΔF/F was very similar to actual ΔF/F in Fam, while in New, the fit of modeled ΔF/F was much worse (six sample cells, *Figure 3A*). To quantify, we compared modeled ΔF/F to actual ΔF/F in two ways: root mean square (RMS) error, which decreases with better fit, and percent of variance explained ($R^2$), which increases with better fit. Using both measures, we found that model fit was significantly worse in New versus Fam$_{Ave}$ for all 5 days, suggesting that

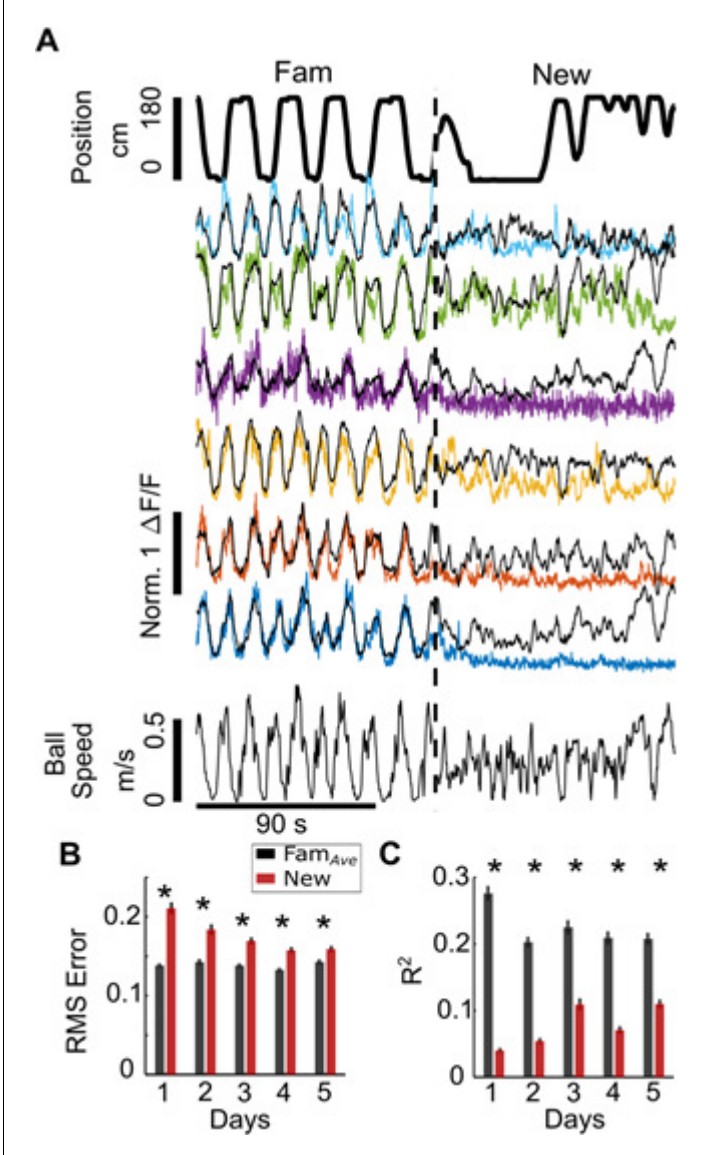

**Figure 3.** Decreased SOM-int activity in New is not explained by altered behavior. (**A**) Gaussian general linear models (GLMs) for individual SOM-ints were trained to predict calcium activity as a function of locomotion, VR movement, and rewards in Fam. In New, modeled ΔF/F (black) is larger than actual ΔF/F (colored traces), indicating that the suppression of activity is greater than that predicted from the model (using example mouse [SOM 5]). Note that mice can move on the ball but not change their VR position, as seen here shortly after transition into New. This occurs when animals 'run' directly into a VR wall so that they are stationary in VR but still moving. (**B**) Model fits are significantly worse in New versus Fam$_{Ave}$ based on average Root Mean Square (RMS) error (lower errors mean better model fit). (**C**) The average amount of variance ($R^2$) capturedby model also shows worse model fit in New (greater $R^2$ means better model fit) (*$p<0.001$ by paired sample t-test Bonferroni-Holm corrections, N = 10, n = 209).

DOI: https://doi.org/10.7554/eLife.47611.013

The following source data and figure supplements are available for figure 3:

**Source data 1.** Statistical tests and results for *Figure 3*.
DOI: https://doi.org/10.7554/eLife.47611.018
**Figure supplement 1.** Behavioral variables are poorly correlated with SOM-int activity in New.
DOI: https://doi.org/10.7554/eLife.47611.014
**Figure supplement 2.** SOM-int GLM performance in different environments.
DOI: https://doi.org/10.7554/eLife.47611.015
*Figure 3 continued on next page*

*Figure 3 continued*
**Figure supplement 3.** Locomotion variables strongly contribute to SOM-int model fits.
DOI: https://doi.org/10.7554/eLife.47611.016
**Figure supplement 4.** Behavioral variables poorly estimate SOM-int activity in New on Day 1.
DOI: https://doi.org/10.7554/eLife.47611.017

changes in behavior, including locomotion, did not explain decreased ΔF/F in New (RMS error, *Figure 3B*; $R^2$, *Figure 3C*).

We also compared model fits in New to Fam and Fam' individually (rather than as $Fam_{Ave}$) and found that model fit was worse in New relative to Fam; however, there was not always a significant difference between the RMS errors of the New and Fam' fits, and this difference became non-significant on Day 2 in New (*Figure 3—figure supplement 2B*). By contrast, $R^2$ values were consistently worse in New vs. Fam' (*Figure 3—figure supplement 2C*). Overall model fits appeared worse in Fam' relative to Fam. This probably results from multiple factors that contribute to worse fits in both Fam' and New. First, there is drift in neuronal representations over time, in this case, the relationship between behavioral variables and cellular activity. Thus predicted activity will tend to be best in data temporally close to the data used to train the model, in this case Fam (we note that we used cross validation where training and test data are independent, although both within the Fam epoch). Second, photobleaching over time contributes to decreased model fit. The model is trained on brighter cells in Fam, resulting in predicted fluorescence changes that are larger than those seen in the same dimmer, photobleached cells later on (photobleaching can be observed in the fluorescence traces shown in *Figure 2—figure supplement 1*). While we attempted to correct for photobleaching using common approaches (such as fitting and correcting with double exponentials), no single solution was appropriate for all cells, primarily because of variability in photobleaching across cells. This confound preferentially impacts RMS error, which is more sensitive to the absolute differences between the actual and predicted fluorescence trace than the $R^2$ value. Finally, there may be a prolonged network effect of exposure to the New environment that crosses into the subsequent Fam' epoch, perhaps due to long lasting neuromodulatory effects. These time-varying factors will also affect model fit in the New epoch, although with less effect because less time has elapsed since model training. These incremental, time-dependent effects appear much weaker than the immediate and drastic loss of model prediction in New.

To evaluate the contribution of individual behavioral variables to cellular activity, we quantified model fits based on single variables (*Figure 3—figure supplement 3*). Overall, models based on all variables had the best fits, while among the individual variables, those representing locomotion (forward and rotational ball speed, and VR speed) contributed the most to model fits of cellular fluorescence. Finally, we investigated the relationship between behavior and neuronal activity by training GLMs in different behavioral epochs. Earlier, we trained models in Fam and asked how well those GLMs predicted activity in New. This assumes that cells have the same relationship between behavior and activity in Fam as in New (otherwise the fit will be poor). By training the GLMs in New, we should achieve a high model fit if there is any predictable relationship between behavior and neuronal activity in New. In fact, GLMs trained on behavioral variables in New were still very poor at predicting activity in New (*Figure 3—figure supplement 4*), whereas in contrast, GLMs trained in either Fam and Fam' predicted activity in familiar environments well, but poorly in New. This suggests a weakened influence of ongoing behavior on SOM-int activity, specifically in new environments. Taken together, these data show that the strong suppression of SOM-int activity in new environments is not simply explained by changes in mouse behavior but is likely to be triggered by contextual novelty itself.

Next, we tested whether soma-targeting PV-ints also show activity suppression in new environments, as shown in real-world experiments (*Frank et al., 2004*; *Nitz and McNaughton, 2004*; *Wilson and McNaughton, 1993*), although not in previous VR experiments (*Hainmueller and Bartos, 2018*; *Sheffield et al., 2017*). Similar to SOM-ints, PV-int somatic calcium activity was also strongly suppressed in New and recovered with repeated exposure to New (*Figure 4*, *Figure 4—figure supplement 1*). Qualitatively, suppression of activity seemed even stronger in PV-ints, perhaps reflecting decreased activity from a higher starting firing rate, as PV-ints generally have greater basal firing rate in vivo (*Royer et al., 2012*; *Varga et al., 2012*). There was also a spectrum of activity

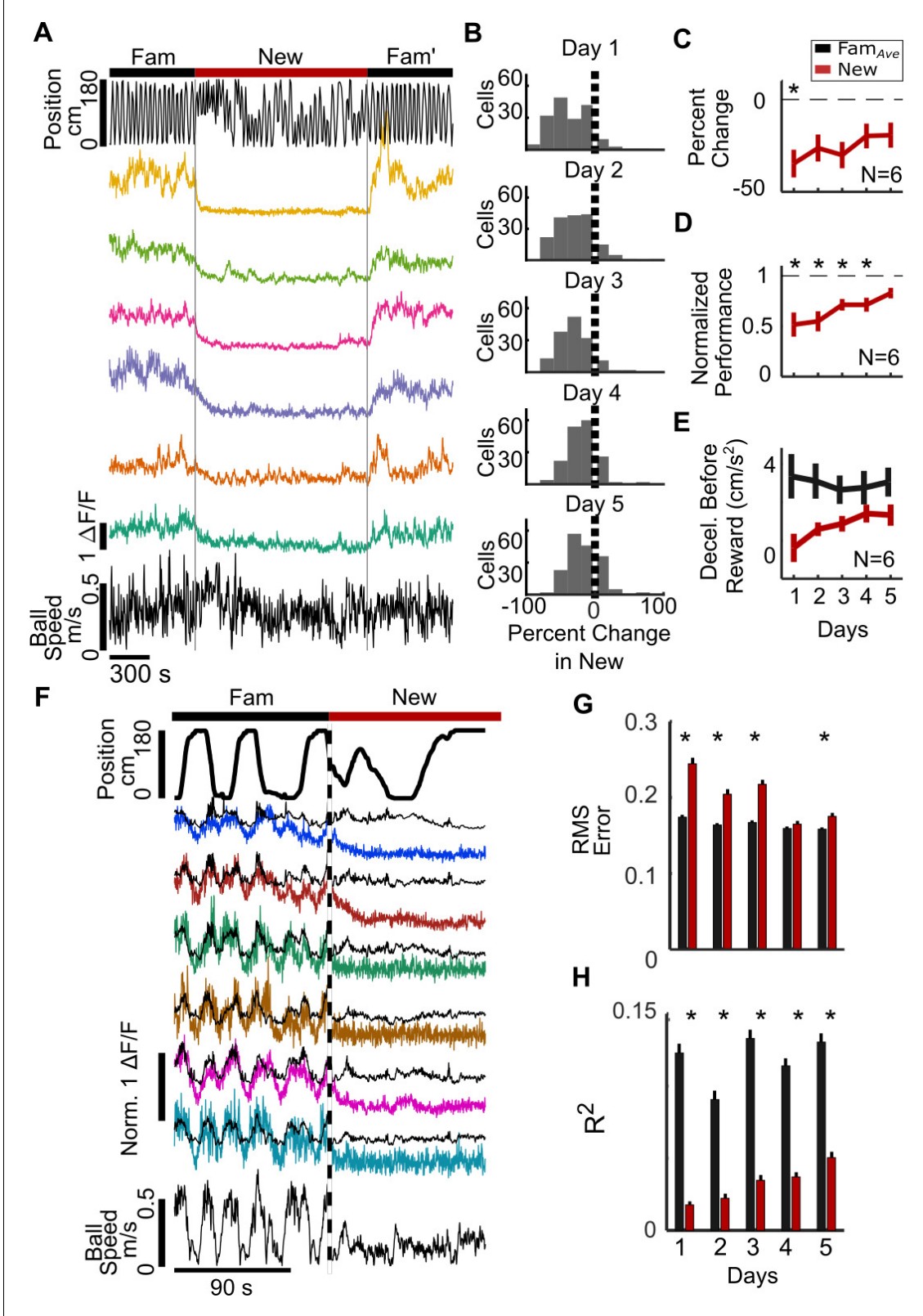

**Figure 4.** PV$^+$ interneuron (PV-int) activity suppression in new environments. (**A**) Example data from an individual mouse (PV 1). Top, position in the VR track of an example mouse. Middle, ΔF/F of sample PV-ints showing activity suppression in New. (**B**) Histogram of percent change in ΔF/F of PV-ints from all mice in New relative to Fam$_{Ave}$ on Day 1, showing initial activity suppression in New that decreases with exposure over days. (**C**) PV-int activity is initially suppressed but recovers over days of exposure to New. (**D**) Performance in a New world increases over days. (**E**) Mice show a non-significant

*Figure 4 continued on next page*

*Figure 4 continued*

trend toward decreased deceleration before reward in New. (**F**) Gaussian general linear models (GLMs) for individual PV-ints were trained as a function of locomotion, VR movement, and rewards in Fam to predict calcium activity. In New, modeled ΔF/F (black) is larger than actual ΔF/F (colored traces), indicating that suppression of activity is greater than that predicted by the model (in example mouse (PV 2)). (**G**) Model fits are significantly worse in New versus Fam$_{Ave}$ based on average Root Mean Square (RMS) error (lower errors mean better model fit). (**H**) Average amount of variance ($R^2$) predicted by model also shows worse model fit in New (greater $R^2$ means better model fit) (*$p<0.001$ by paired sample t-test Bonferroni-Holm corrections, N = 6, n = 172).

DOI: https://doi.org/10.7554/eLife.47611.019

The following source data and figure supplements are available for figure 4:

**Source data 1.** Statistical tests and results for *Figure 4*.
DOI: https://doi.org/10.7554/eLife.47611.026
**Figure supplement 1.** PV-int activity suppression over 5 days of remapping into New.
DOI: https://doi.org/10.7554/eLife.47611.020
**Figure supplement 2.** Broad PV-int firing fields in Fam and New on Day 1.
DOI: https://doi.org/10.7554/eLife.47611.021
**Figure supplement 3.** Suppression of PV-int neurite activity.
DOI: https://doi.org/10.7554/eLife.47611.022
**Figure supplement 4.** Behavioral variables are poorly correlated with PV-int activity in New.
DOI: https://doi.org/10.7554/eLife.47611.023
**Figure supplement 5.** PV-int GLM performance in different environments.
DOI: https://doi.org/10.7554/eLife.47611.024
**Figure supplement 6.** Behavioral variables poorly estimate PV-int activity in New on Day 1.
DOI: https://doi.org/10.7554/eLife.47611.025

suppression in individual PV-ints, ranging from strong suppression to moderate activation in New (*Figure 4B*). Behavioral performance (rew/min) was also impaired in New relative to Fam, although in this batch of mice, performance was impaired over 4 days of exposure to New (*Figure 4D*). Deceleration before stops showed the same trend as SOM$^+$ mice, with initially less deceleration and then increasing deceleration with repeated exposures to New; however, this change in deceleration was not significant (*Figure 4E*, note that N = 6 in this cohort). Using a GLM trained on behavioral data and cell fluorescence in Fam, we found that model fit was significantly worse in New compared to Fam$_{Ave}$ (*Figure 4F,G,H*), suggesting that changes in behavior were not responsible for decreased cellular activity.

Firing fields of PV-ints were broad in both environments (*Figure 4—figure supplement 2*) and, similar to cell somata, PV-int neurites were suppressed in New (*Figure 4—figure supplement 3*). To investigate the relationship between behavior and PV-int activity in Fam and New, we first correlated these variables with neuronal activity, again finding relatively high correlation between locomotion and activity in Fam and Fam′, but markedly decreased correlation in New (*Figure 4—figure supplement 4*). GLMs trained on behavioral data in Fam and PV-int activity predicted significantly better neuronal activity in Fam and Fam′ than in New (*Figure 4—figure supplement 5*), although again GLM model fits were better in Fam than in Fam′, likely for the same reasons described for SOM-ints. To further investigate the role of behavior on PV-int activity in New, we trained GLMs in New, rather than in Fam. Similar to SOM-ints, GLMS trained in New were poor at predicting activity in New (*Figure 4—figure supplement 6*), whereas GLMs trained in either Fam or Fam′ predicted activity in familiar environments well, but again poorly in New.

Overall, the results from PV-ints were similar to those of SOM-ints when exposed to new environments, with strong suppression of activity that recovered with repeated exposure and paralleled the recovery of task performance. Modeling of activity also suggests that changes in behavior in New were not responsible for decreased activity. Taken together, these data from PV- and SOM-ints show that two major classes of inhibitory neurons, targeting distinct subcellular targets, are inhibited in new environments, consistent with a critical role for inhibition suppression in the reorganization of network activity in new environments.

Another potential interpretation of interneuron activity suppression in new environments is that suppression is driven by surprise at the context switch, with habituation to this surprise gradually restoring interneuron activity, with no relationship between the return of interneuron activity and the

recovery of task performance in the new context. To test this possibility, we dissociated surprise at the context switch from task performance in the new environment by replacing the New environment with a no-task, no-reward epoch and a static visual scene (black screen). Under these conditions, if surprise drives interneuron activity suppression and activity recovery is due to habituation, we would see the same suppression and recovery over time as previously shown. On the other hand, if performance recovery is necessary for the restoration of interneuron activity, we should see sustained inhibitory suppression.

Taking advantage of the long-term recording stability of two-photon calcium imaging, we recorded the same SOM-int cells from a subset of the mice (N = 6) used in the remapping experiment, allowing us to compare directly the kinetics of activity recovery when performance recovery was present or absent. The 'No Task' environment evoked strong suppression of SOM-int activity, similar to the suppression seen when mice are switched into New (*Figure 5A–D*). However, over 5 days of exposure to the same 'No Task' environment, activity remained strongly suppressed (*Figure 5E*, *Figure 5—figure supplement 1*). In marked contrast, the same cells from the same mice showed strong recovery of activity over 5 days of exposure to New (*Figure 5E*, showing data from six mice used in *Figure 2* that were also, later, switched into the 'No Task' environment). Thus, recovery of SOM-int activity is unlikely to be the result of habituation to surprise and may rather depend on the recovery of task performance in the new context.

Our previous analyses examined the activity of individual interneurons. Next, we asked how interneuron ensemble activity was altered in our experiments. Here, we measured short-time scale population correlation by calculating all pairwise cell-cell activity correlations within 5 s non-overlapping time bins, averaging those values to get a mean activity correlation for that time bin, then averaging across all mice. The resulting correlation value reports how similar activity is across all cells for that time bin, and the time series of all values shows how this short-time population correlation measure evolves over time. SOM-ints show stable short-time population correlation in Fam that drops significantly on the first day of New, with a return to high short-time population correlation on return to Fam'. By the second day of New, population activity shows similar levels of correlation across Fam, New, and Fam' (*Figure 5—figure supplement 2A,B*). In contrast, SOM-ints in 'No Task' periods show low short-time population correlation, which remains depressed throughout 5 days of exposure (*Figure 5—figure supplement 2C,D*). PV-ints in new environments also showed loss of short-time population correlation in New relative to Fam, although unlike SOM-ints, this decrease was significant for all 5 days (*Figure 5—figure supplement 2E,F*). These results show that interneuron activity at the population level is less similar in new environments than in familiar environments, likely due to a combination of suppression of activity and decreased influence of locomotion on activity.

## Consistent inhibitory structure across new contexts

On average, SOM-ints are suppressed in a new environment (*Figure 2E*), but the degree of activity suppression is heterogeneous across neurons, ranging from strong inhibition to moderate activation in individual cells (*Figure 2D*). It is clear that interneurons respond differently in new environments, raising the question of whether individual interneurons show different activity dynamics in different virtual worlds, similar to the way that hippocampal pyramidal neurons stochastically remap. Alternatively, are these activity dynamics context-invariant, suggesting a consistent network role for individual interneurons?

We tested whether the structure of SOM-int activity suppression was stochastic or consistent by putting a subset of the animals previously described through a second remapping protocol, in which they were exposed to another distinct and novel visual virtual environment, labeled 'New 2,' with the original novel environment now labeled 'New 1' (performance recovery is similar between the two New environments, *Figure 1—figure supplement 1G*). By recording the same cells across the two remapping protocols, we could correlate the magnitude of each cell's activity suppression in New 1 vs. New 2. If SOM-ints are stochastically recruited by network activity, there should be no correlation in activity suppression across New 1 vs. New 2. To the contrary, we found strong correlation between activity suppression in New 1 vs. New 2 in individual SOM-ints. This correlation was strong on Day 1, and strikingly, this correlation was significant across all days of the remapping protocol (*Figure 6A*, Day 1, p<0.001; Day 2, p<0.001; Day 3, p<0.001; Day 5, p<0.001). We, and others, have previously verified place cell global remapping across different virtual environments (*Arriaga and Han, 2017*; *Gauthier and Tank, 2018*; *Hainmueller and Bartos, 2018*;

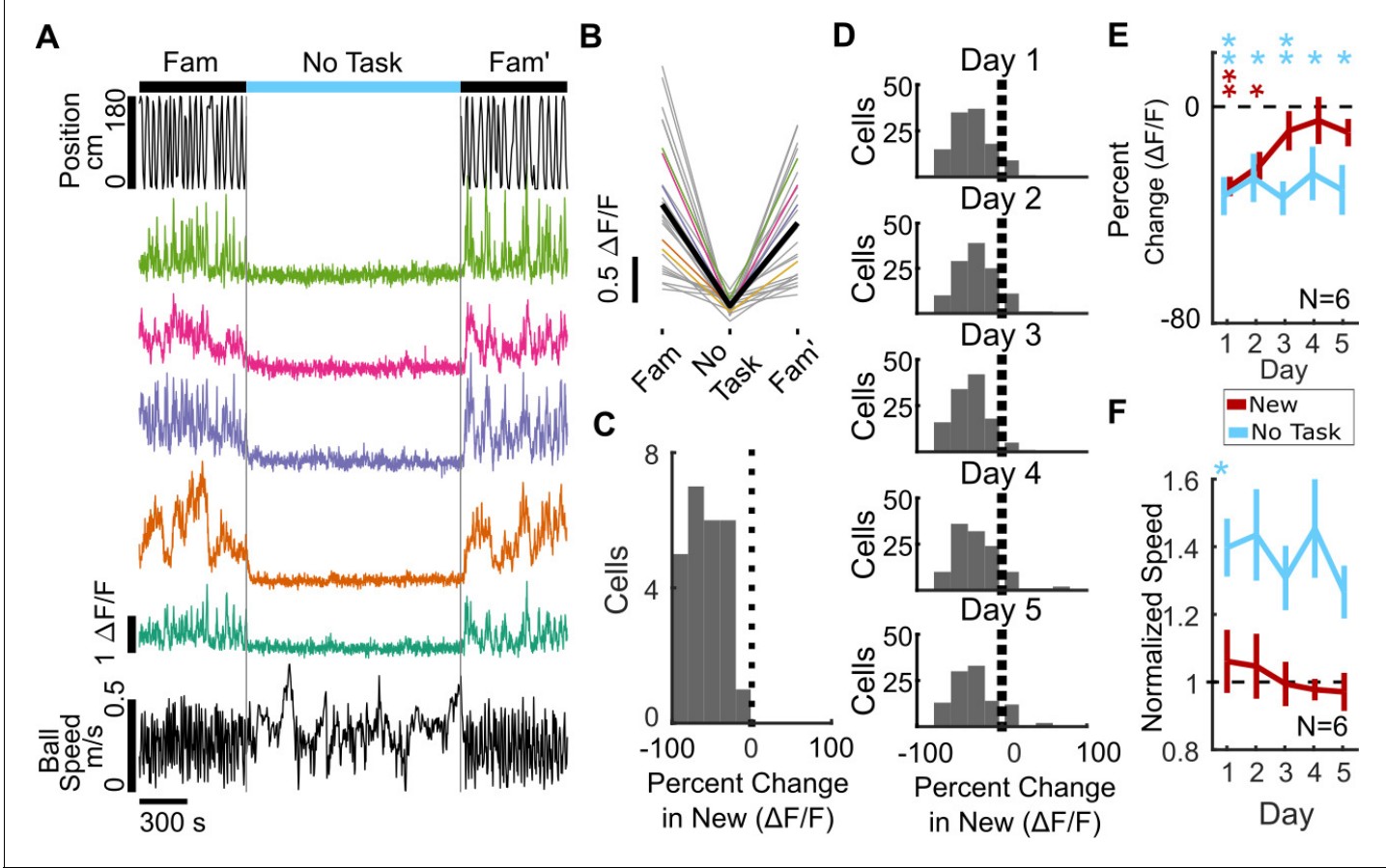

**Figure 5.** SOM-int activity suppression remains high when performance recovery is blocked in a 'No Task' environment. (**A–C**) Example data from an individual mouse (SOM 6). (**A**) Cells from the sample mouse are strongly suppressed during the 'No Task' epoch (static black screen, no rewards). Top, position in VR track, middle, ΔF/F of sample cells, and bottom, ball speed. (**B**) ΔF/F of all cells from an example mouse on Day 1 of 'No Task' exposure showing activity suppression (mean of all cells in black). (**C**) Histogram of percent change in ΔF/F of SOM-ints from the example mouse on Day 1 of 'No Task' showing strong suppression. (n = 18). (**D**) Interneurons remain suppressed over several days of 'No Task' exposure. Histogram of percent change of all cells. (**E**) In 'No Task' exposure, SOM-ints remain suppressed in contrast to recovery during exposure to New. ( The same six mice, which are a subset of the ten mice used in *Figure 2E*, were exposed to No Task and New). (**F**) Average speed in 'No Task' environment increases relative to Familiar, in contrast to New. (n.s. p>0.05, *p<0.05, **p<0.01, ***p<0.001 by paired sample t-test or one-sample t-test with Bonferroni-Holm corrections, N = 6, n = 116).

DOI: https://doi.org/10.7554/eLife.47611.027

The following source data and figure supplements are available for figure 5:

**Source data 1.** Statistical tests and results for *Figure 5*.
DOI: https://doi.org/10.7554/eLife.47611.030

**Figure supplement 1.** SOM-int activity suppression in the No Task environment.
DOI: https://doi.org/10.7554/eLife.47611.028

**Figure supplement 2.** Short-time correlation declines in new environments.
DOI: https://doi.org/10.7554/eLife.47611.029

*Sheffield et al., 2017*), strongly suggesting that this consistent functional inhibitory network structure occurs despite differing ensembles of activated pyramidal neurons.

A trivial explanation for these results could be that this correlation results from general similarities across the two behavioral epochs, such as having equivalent tasks or a shared layout of the virtual worlds. To probe the structure of SOM-int activity suppression in a drastically different context, we used the 'No Task' epoch described previously (*Figure 5*). Here, the visual scene is distinct and static, and there is no behavioral task. Even here, when comparing activity suppression in New 1 vs. 'No Task' epochs, we found significant correlation for each cell on Day 1, indicating that the

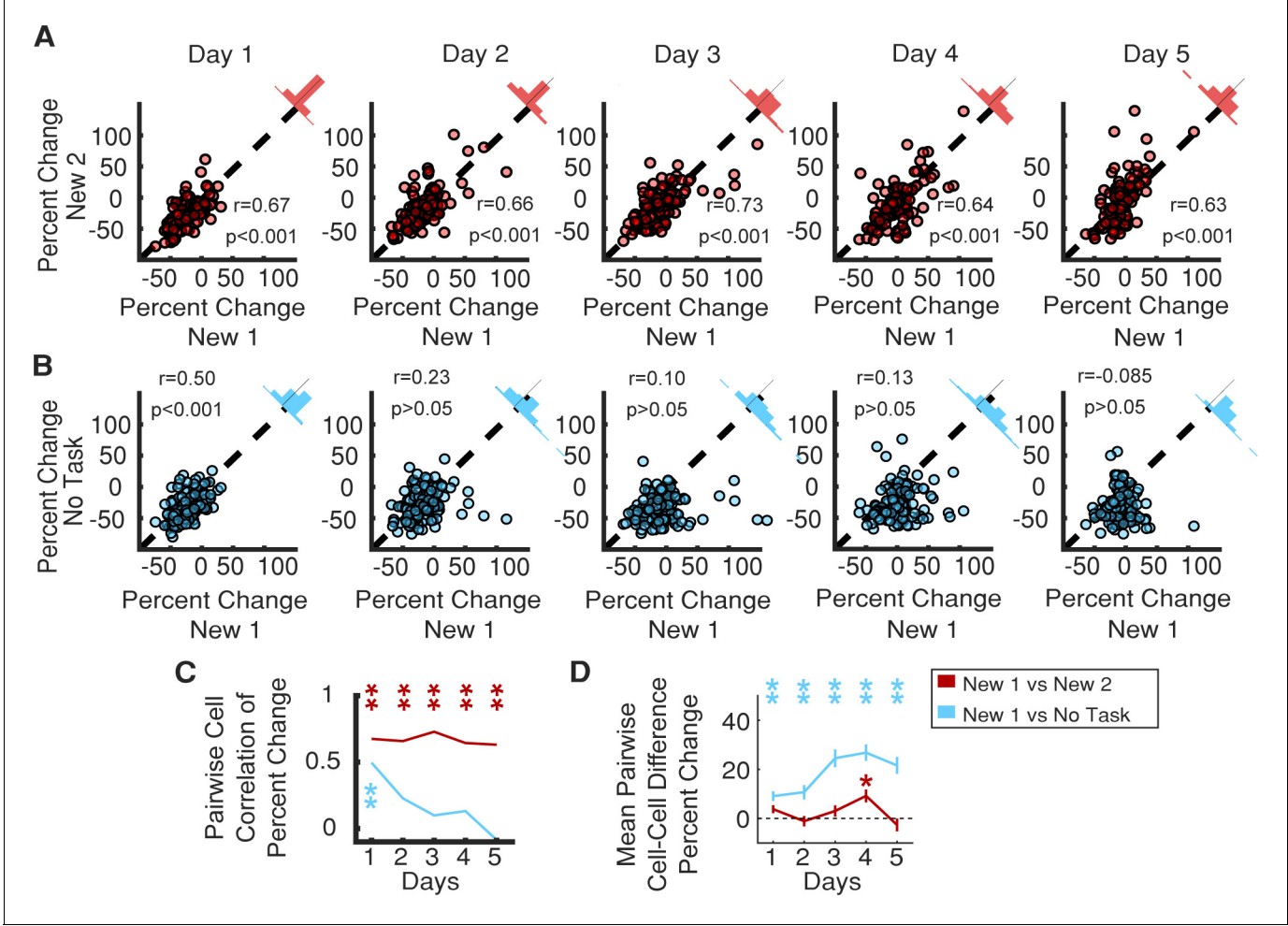

**Figure 6.** Consistent SOM-int activity responses across different new environments and in 'No Task' epoch. (A) Individual SOM-ints show correlation in activity modulation in two distinct New environments. (B) Similar correlation of activity modulation is seen between New 1 and 'No Task' exposures for Day 1. On subsequent days, correlation disappears as SOM-int activity begins to return in New 1 while remaining suppressed in 'No Task'. (C) Summary of correlation data from (A) and (B). Correlation between percent change of cells between two remapping sessions or between remapping session one and the 'No Task' exposure session. (D) Mean difference in percent change in activity in cells between remapping and 'No Task' exposure settings (*p<0.01, **p<0.001, one-sample t-test with Bonferroni-Holm correction N = 6, n = 116).

DOI: https://doi.org/10.7554/eLife.47611.031

The following source data and figure supplements are available for figure 6:

**Source data 1.** Statistical tests and results for *Figure 6*.
DOI: https://doi.org/10.7554/eLife.47611.035

**Figure supplement 1.** Characterization of suppressed SOM-ints.
DOI: https://doi.org/10.7554/eLife.47611.032

**Figure supplement 2.** SOM-int network activity structure is stable across environments.
DOI: https://doi.org/10.7554/eLife.47611.033

**Figure supplement 3.** SOM-int network activity structure is stable across days in the same environment.
DOI: https://doi.org/10.7554/eLife.47611.034

functional inhibitory network structure for these two very different behavioral epochs is very similar. We also measured correlation in activity suppression throughout the 5-day protocol. Although the correlation was significant on Day 1 (*Figure 6B*, p<0.001), it was not on Days 2–5. This was not surprising because inhibitory activity in New 1 recovers with increasing task performance, whereas suppression remains strong in 'No Task' epochs.

The rapid time-dependent loss of correlation between the New 1 and 'No Task' epochs reinforces how striking the correlation structure is for New 1 vs. New 2, indicating not only that initial suppression is correlated but that there is also consistency in the temporal dynamics of this structure in time across 5 days (*Figure 6C*). Similarly, the mean difference in percent change for each cell between New 1 and New 2 remains stable across the remapping paradigm, whereas the difference between New 1 and the 'No Task' epoch steadily increases over the 5-day course of exposure to each environment (*Figure 6D*).

These results show that each cell exhibits a consistent degree of activity suppression across multiple new environments. To understand more about how this structure arises, we looked for other factors that were associated with each cell's magnitude of activity suppression. First, we examined the recovery of activity in cells as a function of initial level of activity suppression on Day 1 of remapping into New 1. Cells stratified by the magnitude of their initial suppression continue to be stratified by activity suppression across the 5-day remapping protocol, with the most strongly suppressed on Day 1 remaining the most suppressed on Day 5, while the least suppressed remain the least suppressed (*Figure 6—figure supplement 1A*). This finding indicates that each cell's initial activity suppression is predictive of future activity throughout the protocol, suggesting that neurons may not be drawn from the same population of functionally homogeneous interneurons. Similarly, cells that were most suppressed in the 'No Task' epoch remained the most suppressed through subsequent days of exposure to this epoch (*Figure 6—figure supplement 1B*).

Does the degree of activity suppression differentially map onto distinct SOM-int cell types? SOM-ints labeled by cre-driver lines are primarily composed of two functionally and anatomically distinct types, OLM and bistratified interneurons (*Klausberger et al., 2004*; *Royer et al., 2012*). The somata of OLM neurons lie in *stratum oriens* (SO) extending into *stratum pyramidale* (SP), whereas bistratified interneurons are mostly in SP. We found no difference in activity suppression between cells with somata in SO vs. SP (*Figure 6—figure supplement 1C*, p=0.38), suggesting that each cell's magnitude of activity suppression was not defined by cell-type, at least at the level of classification of OLM and bistratified interneurons. It remains possible that even more specific cell-type classification, perhaps through single-cell sequencing methods, may identify such a distinction (*Harris et al., 2018*). We also found that baseline fluorescence (which could be indicative of basal firing rate, with strong caveats) exhibited no relationship to activity suppression (*Figure 6—figure supplement 1D*). There was no relationship between soma area and activity suppression (*Figure 6—figure supplement 1E*, p>0.99).

We previously identified two distinct populations of SOM-ints, one whose activity was positively correlated with locomotion and another whose activity was anti-correlated (*Arriaga and Han, 2017*). These two populations, as measured by the phase angle of the hilbert transform of the cell's correlation between stop-triggered mean activity and running speed, were also present in these data. However, there was no difference in activity suppression between the two (*Figure 6—figure supplement 1F*, p>0.99). Thus, the degree of activity suppression was not readily explained by these previously identified cell classes or cellular properties.

Across the cell population, there was variability in the goodness of model fit, so we investigated whether model fit in Fam was predictive of activity suppression in New. Using both RMS error and $R^2$, we found no relationship between the two variables (*Figure 6—figure supplement 1G,H*). It is also possible that activity suppression is locally organized, or otherwise distance-dependent, with nearby cells having similar magnitudes of fluorescence changes in New. However, there was no relationship between cell-cell distance and activity suppression (*Figure 6—figure supplement 1I*). We did, however, find that closer cells have more similar activity, although the relationship was weak (*Figure 6—figure supplement 1J*).

Our results identify stable SOM-int activity dynamics across different environmental contexts. Is this stability restricted to activity dynamics or is the general population activity structure similarly stable? Earlier we found that overall SOM-int population activity becomes less coordinated when comparing average population correlation between familiar and new or 'No Task' environments (*Figure 5—figure supplement 2*). Here we examined population activity structure by calculating cell-cell activity correlations and asking whether these pairwise activity relationships were stable across environments. In an example animal, in the matrix of cell-cell activity comparisons, there is general preservation of activity relationships between cells in Fam and New (*Figure 6—figure supplement 2A,B*). This can also be seen directly in activity traces, in which cells that have similar activity

in Fam also have similar activity in New (*Figure 6—figure supplement 2C*, cells 1 and 2), whereas cells that have dissimilar activity in Fam are also dissimilar in New (cells 4 and 5). Indeed, we found correlation in cell-cell activity when looking at Fam vs. New, New 1 vs. New 2, Fam vs. No Task, and New vs. No Task (*Figure 6—figure supplement 2D–G*). This finding of stable activity structure across environments is seemingly at odds with our earlier finding of decreased short-time population correlation between Fam and New (*Figure 5—figure supplement 2*). However, our measure of short-time population correlation is an average of all correlation values across the population, while activity structure relies on individual cell-pair correlations. Thus on average, short-time population correlation goes down in New (as can be seen by the overall decrease in correlation values in *Figure 6—figure supplement 2B* relative to *Figure 6—figure supplement 2A*), while cell-pair correlations are maintained (*Figure 6—figure supplement 2D–G*).

Having identified overall stability in cell-cell activity correlation in different environments, we asked whether there was a relationship between each pair's correlation level and activity suppression in New. To assign a single activity suppression number to each cell pair, we used the absolute difference between activity suppression values for individual cells; thus pairs with similar activity suppression will have a low difference, whereas pairs with dissimilar suppression will have a larger value. We found that cell pairs with dissimilar amounts of activity suppression in New also had lower activity correlations in Fam (*Figure 6—figure supplement 2H*). In particular, this relationship seemed driven by cell pairs with low activity correlation that also have strongly differing amounts of activity suppression (left side of graph). We also asked whether there was a bias toward higher or lower mean activity suppression on the basis of the cell-pair correlation value. We found that cell pairs with more similar activity had significantly less activity suppression, although this relationship was relatively weak (*Figure 6—figure supplement 2I*). Finally, we found that activity structure based on cell-pair activity correlation was stable across 5 days of exposure in Fam, New, and 'No Task' (*Figure 6—figure supplement 3*). These data identify overall preserved SOM-int activity structure, in addition to consistent activity dynamics, across different environments.

## Discussion

In this work, we addressed critical questions in neuronal network function: how is interneuron activity dynamically regulated during spatial exploration, and is there a persistent structure in the resulting functional interneuron activity dynamics? We found that the activity of SOM- and PV-ints was initially strongly suppressed upon exposure to a novel virtual environment, and that activity recovered as animals learned to adapt a goal-directed spatial navigation task to this new environment. Furthermore for SOM-ints, there was a persistent functional network structure in the transition from familiar to novel environments. Each SOM-int exhibited a characteristic amount of activity suppression in multiple new environments, as well as in a drastically different 'No Task' environment where there was no recovery of task performance.

Our findings are consistent with a model in which entering new environments triggers decreased network inhibition, which then gradually recovers over the course of learning to stabilize the network (*Frank et al., 2004*; *Wilson and McNaughton, 1993*). We found suppression of both dendrite- and soma-targeting interneurons (SOM- and PV-ints, respectively), suggesting general disinhibition of postsynaptic pyramidal neurons during spatial exploration in this task, rather than subregion-specific changes (*Chen et al., 2015*; *Hainmueller and Bartos, 2018*; *Sheffield et al., 2017*). This loss in inhibition may act as a permissive gate for learning by enhancing synaptic plasticity in the pyramidal neurons. However, an important limitation of our data is that, while we measured calcium activity in interneuronal somata and neurites (axonal and dendritic), the actual inhibitory drive onto pyramidal neurons during spatial exploration remains unknown.

We hypothesized that the recovery of interneuron activity in new environments was dependent on task performance recovery and tested this by switching mice into a 'No Task' environment. Indeed SOM-int activity then remained suppressed rather than recovering, consistent with a role for task performance in modulating interneuron activity dynamics. However, although we eliminated task recovery, multiple variables change in the 'No Task' epoch, including the loss of virtual world, water rewards, and clear task rules. These other factors rather than any dependence on behavioral performance, either singly or in combination, could be responsible for the extended suppression of interneuron activity. More controlled manipulations of rewards and task rules in novel VR

environments should be able to isolate the contribution of these variables to the recovery of interneuron activity. One factor that merits more in-depth investigation is the role of task engagement in controlling network activity. It was striking that, in the 'No Task' epoch, the normal positive correlation between locomotion and interneuronal activity was greatly weakened: even though mice ran on the ball at high speed, SOM- and PV-int activity remained low. This is in marked contrast to head-fixed animals running on a track that have never been trained in a VR task, which show a customary positive correlation (*Katona et al., 2014*). In our mice, it is likely that even the *expectation* of some VR world and task is sufficient to engage a state-dependent change in network activity. Thus, the low interneuron activity state may signify a network that is primed for learning, rather than an indication of novelty.

We have shown long-lived SOM- and PV-int suppression that is temporally distinct from transient suppression of SOM- and PV-ints during place cell global remapping (*Nitz and McNaughton, 2004*; *Sheffield et al., 2017*; *Wilson and McNaughton, 1993*). How much interneuron activity suppression is associated with novelty-induced place cell global remapping and how much is due to task performance? In both real world and VR experiments, switching animals to new environments triggers a few minutes of activity suppression and the formation of new place cell maps on the same timescale (*Frank et al., 2004*; *Hainmueller and Bartos, 2018*; *Muller and Kubie, 1987*; *Nitz and McNaughton, 2004*; *Sheffield and Dombeck, 2015*; *Wilson and McNaughton, 1993*). By contrast, in a learning task with no global remapping where freely moving rats learned new goal locations in a familiar environment, fast-spiking putative interneurons both increased and decreased activity as performance increased (*Dupret et al., 2013*). This reflected dissolution of ensembles encoding old reward maps and the formation of new cell assemblies, comprising pyramidal and interneurons, that represent new reward locations. Taken together, it is likely that the initial SOM-int activity suppression in our experiments is triggered by the switch into a new context and that associated global remapping, while slowly increasing interneuron activity thereafter, is associated with additional map refinement related to task performance. Further experiments using more detailed simultaneous recordings of pyramidal and interneuronal activity will refine this picture, whereas more subtle environmental manipulations will help to dissociate global remapping effects from learning.

Another crucial contribution of this study was the discovery of stable inhibitory interneuron activity structure in different environments, which was enabled by our long-term recording of identified cell-types. First, network activity structure (measured as pairwise cell-cell activity) is stable across different contexts, including the drastically different 'No Task' epoch. This suggests that the SOM-int network either continues to receive similar afferent inputs in familiar and new environments, or that network activity structure is locally stabilized, perhaps by gap junctional coupling in the interneuronal network. Second, interneuron activity dynamics were stable on a cell-by-cell basis, with each interneuron having a consistent level of activity suppression, both across multiple new environments and in the 'No Task' epoch. These findings reveal an underlying inhibitory circuit infrastructure that may serve to shape information encoding in the local pyramidal cell network. Future work will investigate whether PV-ints and other interneuron types show similar stability in activity dynamics across multiple environments or whether SOM-ints are unique in this regard.

Our finding of stable interneuron activity dynamics suggests that inhibition may play an active role in encoding information in the network, perhaps by regulating activity in connected ensembles of pyramidal neurons. Pyramidal neurons downstream of strongly suppressed interneurons are more likely to express plasticity, as a direct result of increased activity due to release of inhibition. Conversely pyramidal neurons downstream of less suppressed interneurons will receive relatively normal levels of inhibition, perhaps limiting plasticity. Thus, this functionally diverse, but consistent, inhibitory structure may act as a mechanism to address a fundamental tradeoff in neuronal network function: balancing stability with plasticity (*Abraham and Robins, 2005*; *McClelland et al., 1995*; *McCloskey and Cohen, 1989*). By modulating the functional properties of downstream neurons, inhibition can create plastic and stable pyramidal ensembles, allowing the integration of new information while preserving existing network function. The regulation of specific pyramidal neuron ensembles through interneuron control has not been well-studied (*Rao-Ruiz et al., 2019*); however, intriguing evidence finding distinct pyramidal neuron populations, which code either for learning (engram cells) or for stable place coding over learning, support the existence of hippocampal microcircuits that specialize in plasticity or stability (*Tanaka et al., 2018*).

One requirement of this model is preferential or targeted connectivity in the outputs of SOM-ints onto pyramidal neurons. Such a scenario is at odds with 'pooled' or 'blanket' inhibition, where interneurons make promiscuous and non-selective synapses (*Fino and Yuste, 2011*; *Packer and Yuste, 2011*), but significant evidence exists for preferential connectivity in both the hippocampus and cortex. In the hippocampus, PV-expressing basket cells preferentially inhibit deep pyramidal neurons projecting to the amygdala, while also being more likely to receive excitation from superficial pyramidal neurons or from deep pyramidal neurons projecting to the prefrontal cortex (*Lee et al., 2014*). In the medial entorhinal cortex, cholecystokinin-expressing basket cells selectively target pyramidal neurons that project extra-hippocampally (*Varga et al., 2010*). Furthermore, in the hippocampus, interneurons participate in cell assemblies with pyramidal neurons and can share coding properties such as place fields (*Ego-Stengel and Wilson, 2007*; *Kubie et al., 1990*; *Marshall et al., 2002*). Similarly, functional subnetworks of interneurons and pyramidal neurons have been identified in the cortex (*Khan et al., 2018*; *Najafi et al., 2019*; *Znamenskiy et al., 2018*). Finally, this work identifying the specialization of interneuron function is complemented by evidence of functionally distinct subsets of CA1 pyramidal neurons (*Danielson et al., 2016*; *Graves et al., 2012*; *Mizuseki et al., 2011*; *Soltesz and Losonczy, 2018*).

We identified structured activity dynamics in the functional responses of interneurons as animals adapted a goal-oriented spatial navigation task to novel virtual environments. What mechanisms might generate the activity dynamics and structure within the interneuron population? Neuromodulatory transmitters targeting G-protein coupled receptors are likely to play a significant role. Novelty or arousal produce strong changes in neuromodulation, with sharp increases in acetylcholine (*Acquas et al., 1996*; *Vinck et al., 2015*), norepinephrine (*Sara et al., 1995*), and dopamine (*Kempadoo et al., 2016*; *McNamara et al., 2014*; *Takeuchi et al., 2016*). Differing levels of inhibitory activity suppression could be set by the expression levels of neuromodulatory receptors in each cell. For example, interneurons show markedly divergent responses to acetylcholine depending on the composition and expression of their receptors (*McQuiston and Madison, 1999*). In vivo, hippocampal SOM-ints are activated by air puffs and this activation depends on muscarinic acetylcholine receptor signaling (*Lovett-Barron et al., 2014*; *Schmid et al., 2016*). Furthermore, synapse gain on SOM-ints after fear conditioning depends on cholinergic input, suggesting additional neuromodulatory effects on SOM-int plasticity (*Schmid et al., 2016*). Finally, recent work investigating the mechanisms of locomotion activation and the suppression of PV-ints in the visual cortex identified norepinephrine and acetylcholine, respectively, as critical effectors (*Garcia-Junco-Clemente et al., 2019*).

Another possible mechanism for suppressing interneuron activity is disinhibitory connections from other interneurons targeting SOM-ints. In the hippocampus, this disinhibitory input can come from local VIP, PV, and SOM interneurons (*Francavilla et al., 2015*; *Lovett-Barron et al., 2012*). Indeed, in our experiments, some SOM- and PV-ints were activated in novel environments, although it remains unclear whether these interneurons provide disinhibitory input. VIP interneurons are strongly associated with disinhibition, and previous work showed that these neurons are necessary for hippocampal-dependent learning (*Donato et al., 2013*; *Turi et al., 2019*).

Finally, it is possible that decreased SOM-int activity is inherited from reduced upstream excitatory input. In this case, feed-forward inhibition is driven by EC and CA2/3 input while feed-back excitation is driven by local CA1 neurons. However, during learning or novelty, CA3 and EC pyramidal neurons don't change their firing rates, while CA1 pyramidal neurons increase activity (*Barry et al., 2012*; *Karlsson and Frank, 2008*; *Nitz and McNaughton, 2004*; *Wilson and McNaughton, 1993*), making it unlikely that suppression of inhibitory cell activity is purely a function of reduced excitatory drive.

Our work identifies inhibitory activity dynamics while revealing that individual interneurons have a consistent functional role across multiple new environments. Together, this work and the work of others highlight functional specialization within defined sets of neurons, which may serve to allow efficient incorporation of new information while maintaining overall network stability.

## Materials and methods

### Animals

All experiments were approved by the Washington University Animal Care and Use Committee. Heterozygotes (+/–) from two cre-driver mice lines on a C57Bl/6J genetic background were used to label parvalbumin-expressing and somatostatin-expressing inhibitory interneurons: $SST^{tm2.1(cre)Zjh}$/J (SOM-cre) and $Pvalb^{tm1(cre)Arbr}$/J (PV-cre; Jackson Labs). All imaging data were from SOM-ints while behavioral data in *Figure 1* were from PV- and SOM-ints.

### Viral injections and hippocampal window implantation

Surgical procedures, VR track running behavior, and two-photon imaging have been described previously (*Arriaga and Han, 2017*). Briefly, mice were injected with adeno-associated virus (AAV) at 2–4 months of age. Mice were anesthetized with isoflurane, and a small (0.5 mm) craniotomy was opened above the left cortex. Virus (AAV1.Syn.Flex.GCaMP6f.WPRE.SV40, Penn Vector Core, University of Pennsylvania, $1.71 \times 10^{13}$ genome copies, diluted 1:1–1:4 with PBS,~50 nL total volume) was pressure injected through a beveled micro-pipette targeting CA1 (−1.8 ML, −2.0 AP, −1.3 DV).

After virus injection, mice were water-scheduled for 1–3 weeks and an imaging cannula (2.8 mm diameter) was implanted above the hippocampus by aspirating the overlying cortex. Mice recovered for at least two weeks after surgery before beginning training.

### VR track running behavior

The virtual reality display used a custom-built semi-cylindrical projection screen (1 ft radius) and two rear projectors (Optima 750 ST). The projection screen was positioned ~30 cm in front of the mouse and occupied 180° of the horizontal view, 16° below the horizon and 35° above. VR world brightness was kept low, both to minimize behavioral inhibition and because stray VR light contributes to noise during calcium imaging. Projectors were set at minimal brightness with an additional darkening film applied to the rear projection screen (Gila Glare Control, Smoke). To measure brightness directly, we used a chromameter (Minolta CS100A). On the basis of the lens configuration of the instrument, we could not measure total brightness to compare luminance across environments, but we could measure smaller visual features (diameter ~1.4 cm). The luminance of all features fell between 0.03 and 3 candela/m$^2$, within the mesopic range of visual function, in which mice are frequently behaviorally active (*Denman et al., 2018*).

Mice were head-fixed on a spherical Styrofoam treadmill supported on a cushion of air from a 3D printed base, which allowed free ball rotation with mouse locomotion. Treadmill movement was tracked using a G400 Logitech mouse configured in LabView (National Instruments). The VR environment was rendered using ViRMEn (Virtual Reality Matlab Engine; *Aronov and Tank, 2014*). Mice were trained to run to alternating ends of a linear VR track (180 cm) for 2–5 weeks until they consistently achieved target performance (>2 rewards/min for one week). After training, mice were imaged during exposure to a new visual virtual world. Remapping experiments consisted of 7 min of behavior in the familiar track (Fam), an instantaneous switch to a novel track of the same length with different visual textures and landmarks (New) for 14 min, and then a return to the familiar (Fam') environment for 7 min. This remapping protocol was repeated for five successive days with the same, decreasingly novel, New world. In a subset of animals, a second remapping task was also performed. This task was identical to the first with the exception of a different New environment (New 2). In addition, this same subset of animals was imaged in a 'No Task' session. This session consisted of 7 min of navigation in the familiar track, 14 min of exposure to a dark screen with no rewards, and a return to the initial familiar environment for 7 min.

### Two-photon imaging

Calcium imaging was performed on a Neurolabware laser-scanning two-photon microscope, with the addition of an electric tunable lens (ETL; Optotune, EL-10–30-NIR-LD) and an f=–100 mm offset lens to change axial focal length rapidly. We imaged 4–6 axial planes spanning up to 250 μm in the z-axis at a total frame rate of 31 Hz, resulting in a per plane sampling rate of 5.2 Hz for a six plane recording and 7.8 Hz for a four plane recording. Field of view in *x-y* was 500 × 500 μm. Laser power (at 920 nm) was ~25–50 mW after the objective and was set independently for each plane imaged.

## Data analysis

Data were analyzed using custom programs written in Matlab (RRID:SCR_001622) available at GitHub (*Arriaga and Han, 2019*, https://github.com/Han-Lab-WUSM/MA-scripts; copy archived at https://github.com/elifesciences-publications/MA-scripts). Images were motion-corrected using cross-correlation registration and rigid translation of individual frames. Slow fluctuations in fluorescence were removed from calculations of $\Delta F/F_0$ by calculating $F_0$ using the eighth percentile of fluorescence intensity from a sliding window 300 s around each time point. ROIs were selected using a semi-automated process. Possible ROIs were identified as contiguous regions with SD >1.5 and an area >90 $\mu m^2$. Overlapping ROIs were manually separated, ROIs were redrawn by hand to separate adjacent cells into distinct ROIs. Unresponsive puncta, or those with low signal-to-noise ratios (initially identified as having a skewness of $\Delta F/F$ in the first familiar environment less than 0.3) were dropped from further analysis. When the same cell was recorded in multiple planes, the brightest ROI was used. Neuropil contamination was removed by subtracting a perisomatic fluorescence signal from an annulus between 5 and 20 $\mu m$ from each ROI, excluding any other possible ROIs ($F_{Corrected-ROI} = F_{ROI} - 0.8 * F_{Neuropil}$).

The percent change in the New environment was calculated on each day for each cell as the ratio between the mean fluorescence in the 14 min New world exposure and the mean of the fluorescence from the two 7 min familiar worlds exposures, normalized by the sum of these means: $Percent\ Change = 1 - \frac{\mu_{New}}{\mu_{(Fam+Fam')}}$.

Cell activity correlation with behavioral variables was calculated by taking the maximum value of the cross-correlation of demeaned time series within a 2 s window of lag. We followed the evolution of cell ensemble correlation over time by taking short-time cell-cell correlation of cell activity using the mean Pearson correlation of all pairwise comparisons in non-overlapping 5 s windows.

## Behavior analysis

Ball movement data, sampled at 1 kHz, was downsampled to match the imaging frame rate. All normalized behavioral metrics were normalized by taking the ratio of the metric in the New world to $Fam_{Ave}$ (mean of Fam and Fam'). Task performance was calculated as the rewards received per minute. Speed was calculated as the Euclidean sum of the forward and rotation components of ball velocity. Deceleration was calculated as the first derivative of the forward component of the ball speed during a 3 s window prior to reward. Deceleration before all rewards in the epoch were averaged (ranging from 13 to 30 rewards/epoch on average). The location of trial failure was identified as the distance from the correct destination end zone at which the animal turns around before reaching the end zone.

Lick behavior was detected using a two-transistor lick detection circuit (*Slotnick, 2009*). Individual licks were not resolvable, so lick responses were binned into lick bouts, defined as a period of repeated lick responses with less than 200 ms between repeated lick signals. The lick rate was calculated as the number of these licking bouts per minute. The fraction of 'correct' licks was calculated as the fraction of licking bouts that occur within ± 500 ms of reward delivery (marked by an audible solenoid click to dispense water). The fraction of licks in unrewarded end zones was calculated as the fraction of incorrect, unrewarded, entries into the track end zone that elicited a bout of licking. Mouse location residency was calculated by binning the track in 20 cm bins and by measuring the amount of time that the animals spent in each bin of the track in each environment.

## General linear model of activity

A general linear model was used to estimate fluorescence as a function of the behavioral parameters that are correlated with cell activity. The model predicts fluorescence as the linear combination of weighted, time-lagged behavior components. The lag used for each component was determined by the time of the peak of its cross-correlation with cell activity. Modeling of interneuron fluorescence was done using the *glmfit* function in Matlab with a normal distribution and an identity link function. The oscillatory nature of interneuron fluorescent activity series, without the large transients typical in pyramidal cells, were better fitted using a normal distribution than using the Poisson distribution commonly used in generalized linear models of pyramidal cell activity. Models were trained using fluorescence data from Fam and fitted using the forward and rotation components of ball speed, the timing of rewards, and the position and speed in the virtual reality environment. Behavioral data

were included at a lag of up to 2 s, as determined by the maximum value of the cross-correlation between each parameter and cell activity. Root mean square (RMS) error and model correlation, for models tested in the environment in which they were trained, were calculated using 10-fold cross validation. Successive models were trained on 9/10 of the data set and tested on 1/10 of the data. The average model performance across these ten sessions was used as the performance of the model. Performance in environments other than the one in which the model was trained was calculated by directly applying the model behavioral data from test epochs, and by calculating RMS and model correlation on the entire resultant time series. Correlation was measured as the Pearson correlation between modeled traces and $\Delta F/F$ in each context, and $R^2$ was taken as the square of the Pearson correlation.

## Experimental design and statistical analysis

Behavior data are reported from 14 mice (seven male, seven female). We recorded 209 somatostatin-cre positive cells (mean = $20.9 \pm 5.45$) from ten mice (eight male, two female) across all 5 days of the initial remapping experiment. For the second remapping and 'No Task' paradigms, we recorded from six of these mice (five male, one female), tracking 107 cells across all three contexts. We recorded from 176 parvalbumin-cre cells (mean = $28.67 \pm 15.02$) from six animals (two male, four female).

The significance of normalized data metrics was calculated using one-sample t-tests of mean values. Differences between familiar and new epochs were calculated using paired sample t-tests of mean values, RMS error values were calculated on each cell in each epoch with paired-sample t-tests.

Significance of RMS error and $R^2$ values were calculated on each cell in each epoch with paired-sample t-tests with Bonferroni-Holm corrections. Correlations of percent change across paradigms were calculated using Pearson correlation. Pearson correlations were used to calculate the correlation between percent change in each New world or No Task epoch. All multiple comparisons were corrected with Bonferroni-Holm corrections.

Locomotion and immobility associated interneurons were identified using the phase angle of the cross-correlation of interneuron activity and running speed, as described previously (*Arriaga and Han, 2017*).

Statistical analyses were performed in Matlab.

## Acknowledgements

This work was supported by the McDonnell Centers for Systems Neuroscience, and Cellular and Molecular Neurobiology, and through the Cognitive, Computational, Systems Neuroscience Pathway at Washington University in St. Louis. We thank Martha Bagnall and Suyash Harlalka for comments.

## Additional information

### Funding

| Funder | Grant reference number | Author |
| --- | --- | --- |
| McDonnell Center for Systems Neuroscience | | Edward B Han |
| McDonnell Center for Cellular and Molecular Neurobiology | | Edward B Han |
| Washington University in St. Louis | Cognitive, Computational, Systems Neuroscience Pathway (Graduate Student Fellowship) | Moises Arriaga |

The funders had no role in study design, data collection and interpretation, or the decision to submit the work for publication.

## Author contributions

Moises Arriaga, Resources, Data curation, Software, Formal analysis, Validation, Investigation, Visualization, Methodology, Writing—review and editing; Edward B Han, Conceptualization, Supervision, Investigation, Methodology, Writing—original draft, Writing—review and editing

## Author ORCIDs

Edward B Han  https://orcid.org/0000-0002-1009-2186

## Ethics

Animal experimentation: This study was performed in strict accordance with the recommendations in the Guide for the Care and Use of Laboratory Animals of the National Institutes of Health. All of the animals were handled according to approved institutional animal care and use committee (IACUC) protocols of Washington University (Animal Welfare Assurance # A-3381-01). The protocol was approved by the Washington University School of Medicine IACUC (#20170230). All surgery was performed under isofluorane anesthesia, and every effort was made to minimize suffering.

## Decision letter and Author response

Decision letter https://doi.org/10.7554/eLife.47611.040
Author response https://doi.org/10.7554/eLife.47611.041

# Additional files

## Supplementary files

• Transparent reporting form DOI: https://doi.org/10.7554/eLife.47611.036

## Data availability

Source data are available at https://doi.org/10.5061/dryad.f83kt85.

The following dataset was generated:

| Author(s) | Year | Dataset title | Dataset URL | Database and Identifier |
|-----------|------|---------------|-------------|-------------------------|
| Arriaga M, Han E | 2019 | Data from: Structured Inhibitory Activity Dynamics During Learning | https://doi.org/10.5061/dryad.f83kt85 | Dryad Digital Repository, 10.5061/dryad.f83kt85 |

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
