## [Decision Letter]

Thank you for submitting your article "Structured Inhibitory Activity Dynamics During Learning" for consideration by *eLife*. Your article has been reviewed by three peer reviewers, and the evaluation has been overseen by a Reviewing Editor and Laura Colgin as the Senior Editor. The following individual involved in review of your submission has agreed to reveal their identity: Andre A Fenton (Reviewer #3).

The reviewers have discussed the reviews with one another and the Reviewing Editor has drafted this decision to help you prepare a revised submission.

Summary of major concerns:

1) Clarify the behavioral task and other aspects of the paper.

Regarding the task, the interpretation is not clear (e.g., does it study learning specifically). It also is not clear whether the authors adequately controlled for several potentially important variables, such as bright light. Clarification about locomotion as a variable is also important to address (see comments of reviewers 1 and 2). In the subsection “Consistent inhibitory structure during learning”, the rationale for the experiments is not clear (see comments of reviewer 2).

2) The model is useful but the authors need to address the ability to make predictions about both 'New' as well as 'Fam' (comments of reviewers 1 and 2).

3) Regarding sample sizes and statistics, sample sizes were small. Also, stating the numbers of animals is necessary for each analysis. For statistics, critical values and degrees of freedom should be included. Also, for transparency, raw data should be included, not only normalized values.

Essential revisions:

The paper would be substantially strengthened by addressing specificity of the findings to the somatostatin subtype of interneuron (see comments of reviewer 1). This could be achieved by more experiments to look at another interneuron subtype. However it could be that some data already available or new analyses can address this point. Note that when making the revisions, the authors need to be careful about what this study (and other studies) proved and did not prove about interneurons. For example, activity of interneurons is what was studied here, not inhibition of pyramidal cells (see comments of reviewer 2). Population analyses, including analysis of cell pairs, rather than individual cells, could be an important and welcome addition (reviewer 3).

Reviewer #1:

In this very nice study, the authors tackle the complex problem of untangling changes in the hippocampus during spatial learning. They use 2-photon imaging of genetically identified interneurons in the hippocampus during learning and performance of a well-characterized linear track task, where the animals have to learn to navigate a virtual environment to acquire rewards. Overall, there are some intriguingly novel results, and the authors have given some thought to not only the observational data but also to modeling the behavior and the evolution of neural responses. However, there are several places where a deeper dive into the data will likely yield more satisfying results.

One thread that runs through the data is that the activity of the SOM interneurons is highly correlated with locomotion. It is thus surprising that the authors do not spend more time and effort to dissociate the elements of the task, including locomotion, reward availability, learning trajectory, and familiarity. Although the authors note that mean running speed does not change in the 'New' condition as compared to the 'Fam' condition, the dynamics of running are clearly different. In the 'Fam' condition, the running peaks in between the reward sites and the animals slow down as they get closer to the reward. The SOM interneuron activity follows these dynamics closely. In the 'New' condition, the running is fairly constant around the mean and the SOM interneurons are also constant. The interneurons do decrease in activity, but based on this data it is possible that the SOM cells are merely tied to the acceleration/deceleration dynamics. It would be informative to include a condition where the animals observed familiar and novel environments and received reward but did not locomote.

Likewise, the model results in Figure 3 and the supplement raise as many questions as they answer. The model does a pretty good job of recapitulating the SOM interneuron activity in the 'New' condition, based solely on locomotion, position, and rewards. Indeed, speed and forward motion are the largest influences on the model fits. As noted above, these results suggest that the SOM activity is closely tied to the animal's movement. Finally, the authors do not sufficiently address why the model fails to generate predictions for the Fam' condition that are as good as the predictions for the Fam condition.

Are the occasional peaks in SOM activity during 'New' correlated with some behavioral event?

Are these findings generic to all interneurons, or only SOM cells? What do excitatory neurons do during this learning period?

Introduction, third paragraph: interneuron contributions to learning have indeed been studied previously, just not in hippocampus – see Makino and Komiyama, 2015, etc.

Reviewer #2:

In this study, the authors used in vivo calcium imaging from somatostatin interneurons over multiple days coupled with a virtual reality set up to investigate the learning-related changes in activity of these inhibitory interneurons in a spatial learning task. The authors described a diminution of activity when animals are in a new environment compared to a familiar environment, with a gradual return of activity associated with familiarization to the new environment. In contrast, in animals surrounded by black screens that can locomote freely with no specific task, somatostatin interneuron activity is also massively reduced during the first exposures and the suppression is maintained during several days, suggesting that the return of activity in the new environment is tied with the re-learning of goal locations instead of a context habituation.

Globally the manuscript is well-written, figures are easy to read and interpret, and the findings are interesting. The paper provides novel information about detailed activity of an identified hippocampal interneuron type during rewarded navigation in a virtual environment, as well as activity changes with exposure to the task in a novel environment, a task analogous to "remapping". Although the findings appear correct and convincing, I find that some conclusion statements are overstatements, particularly about re-learning the goal location task and the "no task" learning. In addition, several points need to be addressed by the authors.

1) The main point of the study is to report activity of SOM-int. However, throughout the paper the authors interchangeably use the term inhibition instead of SOM-int activity. Of course SOM-ints are inhibitory interneurons but since the authors never assessed if the SOM-int activity resulted in inhibition of pyramidal cells, it is incorrect to refer to inhibition (or inhibitory activity), instead of inhibitory cell activity. This needs to be corrected throughout the manuscript (Abstract, Introduction (last paragraph), Results (subsection “SOM-int activity suppression during learning in new environment”, second and fifth paragraphs), Discussion (first paragraph)).

2) It is not very clear what kind of learning is occurring in the new environment task. The animal is trained first to run across a visual-cued track (fam env) to obtain reward at the end, then turn around and repeat in the other direction. This task takes weeks to learn. Then switching to the new environment, the animal must keep running to each end of the new environment for reward. So the learned behavior is running to end, which is not an ideal spatial learning task. Also in the new environment, the animal must learn to run to the end of the new environment, regardless of visual cues. It is not clear to me if as stated by the authors "mice need to re-learn the task with new visual cues" in the new environment. Rather it seems the same task in a new visual cued environment (remapping?). In fact the performance of the mice in the new environment is significantly different from that in the fam environment, only for the first exposure. Over the next 2-5 days, most performance measures (Figure 1 and Figure 1—figure supplement 1; most importantly correct licks) are not different from the familiar environment, except for deceleration before reward. In addition, the changes in cell activity in new environment (Figure 2E-H) are also significantly different for day 1 only. To me this is not an unambiguous learning task, but a change of context in which the animal adapts the initial running task. Thus, the authors must change their interpretation of the results regarding learning in the new environment (Abstract, end of Introduction, Results and Discussion).

3) Authors state in the text that they used a "large cohort of mice" to characterize behavioral performances and use a subset to perform somatostatin interneurons imaging. The authors report only 9 mice (5 male, 4 female) for behavioral data and 8 mice (6 male, 2 female) for SOM-int imaging which is NOT a subset (at least concerning male mice). This needs clarification. Also 9 mice is not a large cohort for behavioral experiments, so this should be corrected. It would be nice to put number of animals used in each experiment in the text core to avoid back and forth consultation with the material section.

4) A lot of plots in this study have normalized values. It would be interesting to communicate more about raw values, even if effects are not as impressive as with normalized representation, just to get an idea of mice mean performances (Figure 1C, E, Figure 1—figure supplement 1E; Figure 2F, H; Figure 4F). Also, Figure 1D shows a difference in mean deceleration before a reward in a new and a fam environment without specifying the average speed of animals or number of rewards used to calculate the deceleration vector. Also it appears that the example given in Figure 1B is not representative. The performance in new environment appears much worse than the group data (1C). A more representative example would be relevant.

5) Another concern is the method used in the Figure 3 to answer a question raised by the authors. They recognize that the effect they observed could be linked with the locomotion of the animal as they described locomotion-correlated and anti-correlated Som-int activity in their previous work (Arriaga and Han, 2017). To address this question, they used a General Linear Model, that is a useful and supportive tool, but it should be considered an additional analysis of their data. Indeed, one would expect the analysis to show the direct correlation between the locomotion or speed of the animal and cell activity. It is one of the most important correlation to establish in association with the GLM. Another concern is the result (Figure 3—figure supplement 1) that the model established in the Fam situation, does not predict well the cell activity in the Fam' (thru day 1-5) when behavior returned similar to Fam. This seems to indicate that the model is not a good predictor of fluo/activity as a function of various VR and behavioral parameters.

6) Another major concern is about the brightness of screens. One of the drawbacks of virtual reality set up is that they are often bright which could be stressful for the animal and also provide artefacts in imaging. There is some concern about the new environment used in this study because a pattern in the track is clearly darker than other patterns. Could the drastic change of behavior or/and activity seen in this new environment be linked with this darker part VR? For the behavior shown in Figure 1B, do the animal spent more time in the darker part? Indeed the return of activity in New environment experiment is also linked with an improvement of performance in this last one, so with more exposure of the brighter part. In accordance with that, Figure 4 shows a drastic suppression of activity during black screens used in the "no task" exposure. It is thus important to check if the behavioral changes and the suppression effect are still there in a new environment that is equally bright (as the Fam). In addition, concerning the "no task" experiment, it is important to check cell activity with animals placed either with fixed lit screens (for example a fixed visual scene of a maze), or in a new environment with no goal (that is not excluding a learning of the environment but separate the goal directed linked activity).

7) Subsection “Consistent inhibitory structure during learning”, first paragraph. It is not clear to me how the two explanations given by the authors can account for the spectrum of Som-int activity modulation. The first explanation of equipotential interneuron hypothesis is quite unlikely and the second of different interneurons with distinct functional roles is most likely, given the body of literature on interneuron specialization. Also it is not evident how the experiments with a 2nd new environment was a test of the 2 explanations. The experiments are very interesting, but another rationale must be given for them.

Reviewer #3:

This manuscript can be an important contribution to the literature. The authors demonstrate a cell-specific tendency of SOM+ interneurons in hippocampus to decrease their activity in novel environments and that the activity returns across time/exposures to the environment in parallel with learning, not with mere familiarity with the novel environment. The manuscript is clearly written and the data clearly presented. I have a few requests that are intended to improve the already strong paper.

1) The "no-task" control is used to dissociate the confounded relationships between novelty and/or learning on the one hand, and the activity suppression of interest, on the other. It also serves to demonstrate the activity suppression is not due to reduced locomotion. However, it should be acknowledged that the no-task condition changed more than one variable; it removed a goal-directed task component as intended, but it also removed a number of other variables such as consumption of water, visual stimuli, exposure to locomotion-contingent changes in sensation, sensorimotor integration. Any subset or weighted-combination of these changes could have contributed to the initial or maintained no-task activity decrease, not only the exclusion of the conditioned learning component.

Perhaps a better or additional control would have been a new environment with yoked-reward (i.e. reward delivered in a spatially random schedule temporally equivalent to the Fam rewarded trials). This would have removed the ability to learn the location of reward but kept all other physical variables the same. Since such unreinforced exploration of an environment is standard, perhaps the authors have relevant data on activity of these cells in such a condition. In any case the authors need to acknowledge the confound and explain the reasons one should or should not take this confound to be seriously.

Figure 3 shows that the position soon after the change to New is invariant at "0" but the ball is moving. The authors need to explain how that can be.

The one thing I am disappointed by in this paper is the authors did not take real advantage the fact that they recorded many cells at the same time. Almost all analyses performed could have been done if the cells were recorded one at a time across the exposures in 69 or more separate mice. Would population/ensemble analyses have led to the same conclusions without adding anything more? As it is there are important unanswered questions that are contained in the data. For example, to what extent are the within-session fluctuations of dF/F activity correlated across the population of cells? From the presented single cell data, I imagine that the activity correlation structure across the population is stable both within and between exposures and that this even maintains in the New and No-task conditions. However, this need not be the case even though on average, activity decreased across the population. A reader is left wondering. Consistent with their speculative conceptual framework that the cells form a structured inhibitory network (setting an inhibitory infrastructure for activity within the excitatory cell population), the authors should examine how short-time scale population activity vectors evolve within and across the exposures. To evaluate and bolster their conceptual framework, they should also examine if the population of cell-pair activity correlations maintains within and across the exposures. Analysis of these between-cell activity relationships would speak directly to the conceptual framework that the authors promote in the Discussion.

Following on from the preceding comment on lack of between-cell activity relationship investigations, the authors could also determine if the topographic proximity between cell pairs has a role in the inter-cell activity relationships. As mentioned above, these additional analyses would empirically help to address some of the issues that the authors raise in their adroit speculations in the Discussion.

---

## [Author Response]

All comments have been addressed and we begin with a summary of major experimental and analysis revisions, with direct point-by-point responses below.

Summary of major revisions

1) To determine if other interneuron classes show similar activity dynamics as somatostatin-expressing interneurons (SOM-ints), we measured calcium activity from parvalbumin-expressing interneurons (PV-ints) during exposure to novel virtual environments, now detailed in Figure 4 and Figure 4—figure supplements 1-6. We confirm similar suppression of activity in new environments in PV-ints.

2) To investigate population dynamics, we examined short-time scale population correlation across different environments (Fam, New, and Fam’) and found decreased population activity similarity between Fam and New that gradually recovered with repeated exposures (Figure 5—figure supplement 2). This loss of population coordination was likely due to a combination of activity suppression and decreased coupling of locomotion to interneuron activity in New.

We examined network activity structure using pairwise cell-cell SOM-int activity correlations. We found these correlations were largely preserved across different environments, including the drastically different “No Task” epoch, indicating persistent activity structure in the interneuron network. (Figure 6—figure supplement 2, 3). This finding of stable activity structure complements and expands on our finding of consistent activity dynamics in new environments.

3) In addition to above changes, we added six new figure supplements and additional panels featuring requested raw data and additional analyses of behavior, calcium activity, and cellular/behavioral correlates of neuronal activity (further detailed below).

4) We added additional mice to existing experiments, four mice to virtual reality (VR) behavior in Figure 1 (N=14) and two mice to SOM-ints in current Figures 2 and 3 (N=10), and Figures 5 and 6 (N=6).

Summary of major concerns:1) Clarify the behavioral task and other aspects of the paper.Regarding the task, the interpretation is not clear (e.g., does it study learning specifically).

We take the reviewer’s point that the task (running to alternating ends of a virtual track in new visual environments) is not solely a learning task and quantification of task performance includes multiple variables such as surprise and motor performance (subsection “Virtual reality (VR) behavior”, seventh paragraph). As such, we now avoid the unqualified use of “learning” throughout the paper, except when describing other studies, during speculation, or with specific qualifications. Generally we agree with the statement from reviewer 2 that mice adapt the initial behavior to new environments and we use the specific construction “learning to adapt the task” (e.g. Abstract). The amount and type of learning that is required for this “adaptation” remains unclear (discussed in the aforementioned paragraph) and bears further investigation. Most importantly we have de-emphasized the usage of “learning” throughout the paper to avoid implying any causal relationship between learning and the cellular activity dynamics reported here.

It also is not clear whether the authors adequately controlled for several potentially important variables, such as bright light.

Regarding brightness in VR as a potential confound, we were careful to minimize environmental brightness, both as a behavioral confound and because excess light contributes to noise during calcium imaging. Projectors were set at minimal brightness with an additional darkening film applied to the rear projection screen (subsection “VR track running behavior”, first paragraph). To directly measure brightness, we used a chromameter. Based on the lens configuration of the instrument, we could not measure total brightness to compare luminance across environments but we could measure smaller visual features. Even the brightest VR features (which constitute a small fraction of the entire visual scene) were 3 cd/m^2^ which is within the mesoscopic range of visual function where mice are frequently behaviorally active (subsection “Virtual reality (VR) behavior”, last paragraph). Furthermore based on track occupancy in different VR worlds, we saw no indication of preferred areas resulting from brightness differences (Figure 1—figure supplement 2). In general, mice were highly motivated to pursue rewards and ran continuously in all worlds, except when consuming reward.

Clarification about locomotion as a variable is also important to address (see comments of reviewers 1 and 2).

We agree that a thorough investigation into the relationship between locomotion and cellular activity is critical and this was the motivation for using general linear models (GLMs) in the original submission. We have supplemented this analysis with additional analyses that more straightforwardly and directly quantify the correlation between various behavioral variables (including locomotion, e.g. forward and rotational components of ball speed, acceleration, etc.) with interneuron calcium activity in familiar and new environments. Similar to the conclusions from our GLM analyses, we found that locomotion variables correlated well with fluctuations in calcium activity in the familiar environment but correlated poorly in the new environment (Figure 3—figure supplement 1, Figure 4—figure supplement 4).

We further investigated the relationship between behavior and neuronal activity by now training GLMs in New. Previously we trained models in Fam (first epoch in the familiar environment) and asked how well those GLMs predicted activity in New (very badly). Here by training the GLMs in New, we do not make the assumption that whatever relationship between behavior and activity that exists in Fam is recapitulated in New. Now if there is any consistent relationship between behavior and neuronal activity in New, modeled fit should be high. In fact, GLMs trained in New poorly predicted neuronal activity in New (Figure 3—figure supplement 4, Figure 4—figure supplement 6).

Taken together, these new analyses further bolster our original conclusion that locomotion is strongly coupled to interneuron activity (both SOM- and PV-ints) in familiar environments, but contributes much more weakly to activity in new environments. The drivers of remaining activity in new environments remain unclear.

In the subsection “Consistent inhibitory structure during learning”, the rationale for the experiments is not clear (see comments of reviewer 2).

The rationale for investigating the consistency of interneuron activity dynamics across different environments is now more simply stated: do interneurons show stochastic activity dynamics in different virtual worlds, similar to the way hippocampal pyramidal neurons stochastically remap, or are their activity dynamics context-invariant, which suggests a consistent network role for individual interneurons (subsection “Consistent inhibitory structure during across new contexts”, first paragraph).

2) The model is useful but the authors need to address the ability to make predictions about both 'New' as well as 'Fam' (comments of reviewers 1 and 2).

Models trained in Fam did not predict activity as well in Fam’ (the return to the original familiar environment after exposure to New) as it did in Fam. This is likely explained by a combination of time-varying factors (subsection “Interneuron activity suppression in new environment”, seventh paragraph).

First, there is drift in neuronal representations over time, in this case, the relationship between behavioral variables and cellular activity. Thus predicted activity will tend to be best in data temporally close to the data used to train the model, in this case Fam (we note that training and test data are independent, although both within the Fam epoch).

Second, photobleaching contributes to decreased model fit. The model is trained on brighter cells in Fam, resulting in predicted fluorescence changes that are larger than those seen in the same dimmer, photobleached cells in Fam’.

Finally there may be a prolonged network effect of exposure to the New environment that crosses into the subsequent Fam’ epoch, perhaps due to long lasting neuromodulatory effects. These time-varying factors will also affect model fit in the New epoch, although with less effect since less time has elapsed since model training. These incremental, time-dependent effects appear much weaker than the immediate and drastic loss of model prediction in New.

3) Regarding sample sizes and statistics, sample sizes were small. Also, stating the numbers of animals is necessary for each analysis. For statistics, critical values and degrees of freedom should be included. Also, for transparency, raw data should be included, not only normalized values.

We added four mice to the overall behavioral cohort and two mice to the SOM-int cohort. We more clearly label the number of animals used in analysis, either directly on the figure or in the figure legend. Critical values and degrees of freedom are now listed in the excel sheet that accompanies all figures. Extensive raw data (activity traces and behavior) have been included from six SOM^+^ (Figure 2: SOM 1, Figure 2—figure supplement 2: SOM 2, Figure 2—figure supplement 1: SOM 1-4, Figure 3: SOM 5, Figure 5, SOM 6) and two PV^+^ mice. Additionally non-normalized performance, speed, and lick rate are displayed in Figure 1—figure supplement 1.

Essential revisions:The paper would be substantially strengthened by addressing specificity of the findings to the somatostatin subtype of interneuron (see comments of reviewer 1). This could be achieved by more experiments to look at another interneuron subtype. However it could be that some data already available or new analyses can address this point.

We have added data investigating the activity dynamics of PV-int in new virtual environments. Indeed we found that PV-ints are profoundly suppressed in New with recovery of activity over repeated exposures. These data now identify suppression of two major classes of inhibitory neurons in novel environments.

We did not address the activity dynamics of PV-ints across multiple new environments in this revision. Preliminary data from PV-ints show some similarity to SOM-ints in this regard, but significant differences. We feel that meaningfully investigating these differences is beyond the scope of this current work. As such, we would prefer to more thoroughly explore these issues in subsequent work.

Note that when making the revisions, the authors need to be careful about what this study (and other studies) proved and did not prove about interneurons. For example, activity of interneurons is what was studied here, not inhibition of pyramidal cells (see comments of reviewer 2).

This is a very important distinction and we have adopted this more precise usage throughout the paper.

Population analyses, including analysis of cell pairs, rather than individual cells, could be an important and welcome addition (reviewer 3).

See Summary of major revisions (#2) above for population analysis revisions.

Reviewer #1:[…] Overall, there are some intriguingly novel results, and the authors have given some thought to not only the observational data but also to modeling the behavior and the evolution of neural responses. However, there are several places where a deeper dive into the data will likely yield more satisfying results.One thread that runs through the data is that the activity of the SOM interneurons is highly correlated with locomotion. It is thus surprising that the authors do not spend more time and effort to dissociate the elements of the task, including locomotion, reward availability, learning trajectory, and familiarity. Although the authors note that mean running speed does not change in the 'New' condition as compared to the 'Fam' condition, the dynamics of running are clearly different. In the 'Fam' condition, the running peaks in between the reward sites and the animals slow down as they get closer to the reward. The SOM interneuron activity follows these dynamics closely. In the 'New' condition, the running is fairly constant around the mean and the SOM interneurons are also constant. The interneurons do decrease in activity, but based on this data it is possible that the SOM cells are merely tied to the acceleration/deceleration dynamics. It would be informative to include a condition where the animals observed familiar and novel environments and received reward but did not locomote.

As noted above, we now directly examine correlation between behavioral variables and cellular activity. We agree that the dynamics between locomotion and activity is critical, which led us to use GLMs to more rigorously and quantitatively model this relationship in individual cells. Acceleration contributed little to the modeled fit of activity (Figure 3—figure supplement 1, Figure 3—figure supplement 4, Figure 4—figure supplement 5, Figure 4—figure supplement 7).

The role of reward is likely central to some of the effects we observe. Dissociating reward from locomotion modulation of interneurons would be highly desirable but preventing head-fixed animals from moving is highly stressful for them. This could be more reasonably accomplished using a different task with trial structure, rather than free running.

Likewise, the model results in Figure 3 and the supplement raise as many questions as they answer. The model does a pretty good job of recapitulating the SOM interneuron activity in the 'New' condition, based solely on locomotion, position, and rewards. Indeed, speed and forward motion are the largest influences on the model fits. As noted above, these results suggest that the SOM activity is closely tied to the animal's movement. Finally, the authors do not sufficiently address why the model fails to generate predictions for the Fam' condition that are as good as the predictions for the Fam condition.

See above, Major Concerns, #1 (Clarificationabout locomotion*…)* and 2.

Are the occasional peaks in SOM activity during 'New' correlated with some behavioral event?

Occasional peaks in SOM-int activity were intriguing but we did not identify any obvious behavioral correlates (such as rewards, etc.). However, future in-depth analysis of behavior or internal states may shed light on their origins.

Are these findings generic to all interneurons, or only SOM cells? What do excitatory neurons do during this learning period?

As mentioned above, PV-ints also show strong activity suppression in new virtual environments and this data is now included in the paper. Preliminary data from our lab shows that a subset of VIP-expressing interneurons is strongly activated in novel VR worlds, consistent with a role in disinhibition. Several published studies (referenced in the paper) have shown that hippocampal pyramidal neurons remap in new virtual environments. Preliminary data show that after initial remapping, pyramidal neuron representations continue to refine over the next few days before stabilizing.

Introduction, third paragraph: interneuron contributions to learning have indeed been studied previously, just not in hippocampus – see Makino and Komiyama, 2015, etc.This is now more clearly stated and these references have been added (Introduction, second paragraph).Reviewer #2:[…] Although the findings appear correct and convincing, I find that some conclusion statements are overstatements, particularly about re-learning the goal location task and the "no task" learning. In addition, several points need to be addressed by the authors.1) The main point of the study is to report activity of SOM-int. However, throughout the paper the authors interchangeably use the term inhibition instead of SOM-int activity. Of course SOM-ints are inhibitory interneurons but since the authors never assessed if the SOM-int activity resulted in inhibition of pyramidal cells, it is incorrect to refer to inhibition (or inhibitory activity), instead of inhibitory cell activity. This needs to be corrected throughout the manuscript (Abstract, Introduction (last paragraph), Results (subsection “SOM-int activity suppression during learning in new environment”, second and fifth paragraphs), Discussion (first paragraph)).We agree and have adopted this more precise usage throughout the paper.2) It is not very clear what kind of learning is occurring in the new environment task. The animal is trained first to run across a visual-cued track (fam env) to obtain reward at the end, then turn around and repeat in the other direction. This task takes weeks to learn. Then switching to the new environment, the animal must keep running to each end of the new environment for reward. So the learned behavior is running to end, which is not an ideal spatial learning task. Also in the new environment, the animal must learn to run to the end of the new environment, regardless of visual cues. It is not clear to me if as stated by the authors "mice need to re-learn the task with new visual cues" in the new environment. Rather it seems the same task in a new visual cued environment (remapping?). In fact the performance of the mice in the new environment is significantly different from that in the fam environment, only for the first exposure. Over the next 2-5 days, most performance measures (Figure 1 and Figure 1—figure supplement 1; most importantly correct licks) are not different from the familiar environment, except for deceleration before reward. In addition, the changes in cell activity in new environment (Figure 2E-H) are also significantly different for day 1 only. To me this is not an unambiguous learning task, but a change of context in which the animal adapts the initial running task. Thus, the authors must change their interpretation of the results regarding learning in the new environment (Abstract, end of Introduction, Results and Discussion).

See above, Major Concerns, #1.

3) Authors state in the text that they used a "large cohort of mice" to characterize behavioral performances and use a subset to perform somatostatin interneurons imaging. The authors report only 9 mice (5 male, 4 female) for behavioral data and 8 mice (6 male, 2 female) for SOM-int imaging which is NOT a subset (at least concerning male mice). This needs clarification. Also 9 mice is not a large cohort for behavioral experiments, so this should be corrected. It would be nice to put number of animals used in each experiment in the text core to avoid back and forth consultation with the material section.We apologize for the poor wording. This phrasing has been changed and additional animals have been added to the behavior cohort and the SOM-int cohort.4) A lot of plots in this study have normalized values. It would be interesting to communicate more about raw values, even if effects are not as impressive as with normalized representation, just to get an idea of mice mean performances (Figure 1C, E, Figure 1—figure supplement 1E; Figure 2F, H; Figure 4F). Also, Figure 1D shows a difference in mean deceleration before a reward in a new and a fam environment without specifying the average speed of animals or number of rewards used to calculate the deceleration vector. Also it appears that the example given in Figure 1B is not representative. The performance in new environment appears much worse than the group data (1C). A more representative example would be relevant.

See above Major Concerns, #3.

Average speed is shown in Figure 1—figure supplement 1F. Deceleration before all rewards in the epoch were averaged (ranging from 13 – 30 rewards/epoch on average) (subsection “Behavior Analysis”).

5) Another concern is the method used in the Figure 3 to answer a question raised by the authors. They recognize that the effect they observed could be linked with the locomotion of the animal as they described locomotion-correlated and anti-correlated Som-int activity in their previous work (Arriaga and Han, 2017). To address this question, they used a General Linear Model, that is a useful and supportive tool, but it should be considered an additional analysis of their data. Indeed, one would expect the analysis to show the direct correlation between the locomotion or speed of the animal and cell activity. It is one of the most important correlation to establish in association with the GLM. Another concern is the result (Figure 3—figure supplement 1) that the model established in the Fam situation, does not predict well the cell activity in the Fam' (thru day 1-5) when behavior returned similar to Fam. This seems to indicate that the model is not a good predictor of fluo/activity as a function of various VR and behavioral parameters.See above, Major Concerns, #1 (Clarification about locomotion…) and 2.6) Another major concern is about the brightness of screens. One of the drawbacks of virtual reality set up is that they are often bright which could be stressful for the animal and also provide artefacts in imaging. There is some concern about the new environment used in this study because a pattern in the track is clearly darker than other patterns. Could the drastic change of behavior or/and activity seen in this new environment be linked with this darker part VR? For the behavior shown in Figure 1B, do the animal spent more time in the darker part? Indeed the return of activity in New environment experiment is also linked with an improvement of performance in this last one, so with more exposure of the brighter part. In accordance with that, Figure 4 shows a drastic suppression of activity during black screens used in the "no task" exposure. It is thus important to check if the behavioral changes and the suppression effect are still there in a new environment that is equally bright (as the Fam). In addition, concerning the "no task" experiment, it is important to check cell activity with animals placed either with fixed lit screens (for example a fixed visual scene of a maze), or in a new environment with no goal (that is not excluding a learning of the environment but separate the goal directed linked activity).

See above, Major Concerns, #1, It also is not clear….

7) Subsection “Consistent inhibitory structure during learning”, first paragraph. It is not clear to me how the two explanations given by the authors can account for the spectrum of Som-int activity modulation. The first explanation of equipotential interneuron hypothesis is quite unlikely and the second of different interneurons with distinct functional roles is most likely, given the body of literature on interneuron specialization. Also it is not evident how the experiments with a 2nd new environment was a test of the 2 explanations. The experiments are very interesting, but another rationale must be given for them.

See above, Major Concerns, #1 (… the rationale.…).

We also emphasize more clearly in the revision that these functionally distinct interneurons are likely from the same subtype of inhibitory interneuron (based on anatomical evidence in Figure 6—figure supplement 1C). While functional specialization of *distinct* interneuron types is well established, little is known about diversity of functional roles *within* the same type.Reviewer #3:[…] I have a few requests that are intended to improve the already strong paper.1) The "no-task" control is used to dissociate the confounded relationships between novelty and/or learning on the one hand, and the activity suppression of interest, on the other. It also serves to demonstrate the activity suppression is not due to reduced locomotion. However, it should be acknowledged that the no-task condition changed more than one variable; it removed a goal-directed task component as intended, but it also removed a number of other variables such as consumption of water, visual stimuli, exposure to locomotion-contingent changes in sensation, sensorimotor integration. Any subset or weighted-combination of these changes could have contributed to the initial or maintained no-task activity decrease, not only the exclusion of the conditioned learning component.Perhaps a better or additional control would have been a new environment with yoked-reward (i.e. reward delivered in a spatially random schedule temporally equivalent to the Fam rewarded trials). This would have removed the ability to learn the location of reward but kept all other physical variables the same. Since such unreinforced exploration of an environment is standard, perhaps the authors have relevant data on activity of these cells in such a condition. In any case the authors need to acknowledge the confound and explain the reasons one should or should not take this confound to be seriously.

While the “No Task” condition provided essential data regarding the stability of activity dynamics of individual interneurons in drastically different environments, we agree that it was not the optimal experiment to tease apart the role of task and rewards from activity modulation. We acknowledge and discuss this in more detail (Discussion, third paragraph).

Figure 3 shows that the position soon after the change to New is invariant at "0" but the ball is moving. The authors need to explain how that can be.

Mice can move on the ball but not change their VR position, as seen in Figure 3 shortly after the transition to New. This occurs when animals “run” directly into a VR wall so they are stationary in VR but still running (Figure 3 legend). This typically does not occur in familiar environments.

The one thing I am disappointed by in this paper is the authors did not take real advantage the fact that they recorded many cells at the same time. Almost all analyses performed could have been done if the cells were recorded one at a time across the exposures in 69 or more separate mice. Would population/ensemble analyses have led to the same conclusions without adding anything more? As it is there are important unanswered questions that are contained in the data. For example, to what extent are the within-session fluctuations of dF/F activity correlated across the population of cells? From the presented single cell data, I imagine that the activity correlation structure across the population is stable both within and between exposures and that this even maintains in the New and No-task conditions. However, this need not be the case even though on average, activity decreased across the population. A reader is left wondering. Consistent with their speculative conceptual framework that the cells form a structured inhibitory network (setting an inhibitory infrastructure for activity within the excitatory cell population), the authors should examine how short-time scale population activity vectors evolve within and across the exposures. To evaluate and bolster their conceptual framework, they should also examine if the population of cell-pair activity correlations maintains within and across the exposures. Analysis of these between-cell activity relationships would speak directly to the conceptual framework that the authors promote in the Discussion.See Summary of Major revisions (#2) above for population analysis revisions.Following on from the preceding comment on lack of between-cell activity relationship investigations, the authors could also determine if the topographic proximity between cell pairs has a role in the inter-cell activity relationships. As mentioned above, these additional analyses would empirically help to address some of the issues that the authors raise in their adroit speculations in the Discussion.We examined the relationship between activity and distance between SOM-int cell somata and found a weak, but significant, relationship where closer cells have greater activity correlation in Fam (Figure 6—figure supplement 1J). The origins or importance of this finding is unclear. There was no relationship between cell pair proximity and similarity in magnitude of activity suppression in New (Figure 6—figure supplement 1I).